

# The polarized sun and sky radiometer SSARA: design, calibration, and application for ground based aerosol remote sensing

Hans Grob, Claudia Emde, Matthias Wiegner, Meinhard Seefeldner, and Bernhard Mayer

Ludwig-Maximilians-Universität, Institut für Meteorologie, Munich, Germany

**Correspondence:** Hans Grob (H.Grob@physik.uni-muenchen.de)

**Abstract.** Recently, polarimetry has been used to enhance classical photometry to infer aerosol optical properties, as polarized radiation contains additional information about the particles. Therefore, we have equipped the SSARA sun and sky photometer with polarizer filters to measure linearly polarized light at 501.5 nm.

We have developed a novel radiometric and polarimetric calibration method, which allows us to simultaneously determine the linear polarizers' diattenuation and relative orientation with high accuracy (0.002 and 0.1°, respectively). Furthermore, we employed a new calibration method for the azimuthal mount capable of correcting the instrument's pointing to within 32 arcmin. So far, this is limited by the accuracy of the sun-tracker. Both these methods are applicable to other sun and sky radiometers, such as Cimel CE318-DP instruments used in AERONET.

During the A-LIFE field campaign in April 2017, SSARA collected 22 days of data. Here, we present two case studies: The first demonstrates the performance of an aerosol retrieval from SSARA observations under partially cloudy conditions. In the other case, a high aerosol load due to a Saharan dust layer was present during otherwise perfect clear sky conditions.

## 1 Introduction

According to the IPCC aerosols have a significant and not entirely understood impact on the earth's climate (IPCC, 2013). In first order, it induces a direct radiative forcing effect. Additionally, it has been established that aerosols have an influence on the development and lifetime of clouds (Albrecht, 1989), which is known as the secondary aerosol effect. In order to study these effects, aerosols have to be retrieved in the vicinity of clouds. To gain insight into processes occuring in or close to the edge of clouds, microphysical properties of the aerosol are required in addition to the total aerosol load, quantified by the aerosol optical depth (AOD). These are, for instance, information about the size distribution of the particles, their index of refraction, and single scattering albedo (related to the absorptance). The combination of these parameters can be used to indentify the chemical composition, and, eventually, source region of the aerosol. This has in turn impact on the aerosol's hygroscopicity, and therefore the microphysical properties of the cloud droplets that might develop from it.




Aerosols can be measured from satellites, and from the ground. While the former has the advantage of global coverage, its spatial resolution is still not good enough to properly resolve smaller clouds and the aerosol in between them. Ground based systems are better suited for these studies, e.g. the AErosol RObotic NETwork (AERONET) that has been established as a large network of ground based sun photometers (Holben et al., 1998; Giles et al., 2019).

Classically, this information is retrieved from multispectral measurements. Recently, polarimetric measurements started to be included as well. Several studies suggest that including polarimetric information in retrievals yields additional information on the aerosol. Xu and Wang (2015) investigated the gain in information content from adding polarized measurements to principal plane and almucantar scans. In a later paper, they applied their retrieval to real-world AERONET measurements (Xu et al., 2015). The retrieval error was significantly reduced for size distribution parameters (50 %), refractive index (10-30 %)
and single scattering albedo (10-40 %). Dubovik et al. (2006) suggest that polarimetric measurements can be used to gain more insight into the aerosol particle shape. This was further examined by Fedarenka et al. (2016), ascertaining an improvement in retrieval stability for fine mode dominated aerosols, and a high sensitivity to particle shape, due to the use of polarimetry.

Predating these efforts was the POLDER instrument abord the PARASOL satellite (Deschamps et al., 1994), measuring polarized reflectance. Its data has been used for aerosol retrievals (Hasekamp and Landgraf, 2007; Hasekamp et al., 2011). More
recently, the Spectropolarimeter for Planetary EXploration (SPEX) has been developed (van Harten et al., 2011). Originally designed as a satellite instrument (van Amerongen et al., 2017), a ground based version has been built. Both of them have been used for retrieving aerosol properties (Di Noia et al., 2015).

Polarimetric instruments require an additional calibration. Prior work on this has been done for polarized CIMEL CE318-DP sun photometers by Li et al. (2010, 2014, 2018). In this paper, we present an alternative approach that overcomes some of their
limitations and reduces the number of required steps by simultaneously determining the polarizers' efficiencies and angles.

Our new methodology was applied to polarized radiance measurements from the SSARA polarized scanning sun and sky radiometer, taken during the A-LIFE (Absorbing aerosol layers in a changing climate: aging, LIFEtime and dynamics) field campaign. It took place in Cyprus during April 2017 and included ground-based components, such as lidar and radar systems, radiometers, and in situ samplers at Paphos and Limassol. Additionally, a research aircraft with in situ instrumentation was
operated from Paphos airport. The goal of the A-LIFE project is to investigate the effects of aerosol on the earth's radiation budget, cloud development and atmospheric dynamics, with a focus on absorbing aerosols, such as black carbon and desert dust. SSARA has previously been employed in the SAMUM-1 and 2, and the SALTRACE field campaigns that had similar goals (Toledano et al., 2009, 2011).

This paper consists of two parts. Section 2 first characterizes the SSARA instrument. Then, it describes the calibration
methods for the instrument and the azimuthal mount. The second part in Sec. 3 introduces the aerosol retrieval, and then presents the findings for two case studies from the A-LIFE campaign. Section 4 summarizes the findings and gives an outlook for further studies. Additionally, a short primer in quaternion algebra is included in Appendix A.





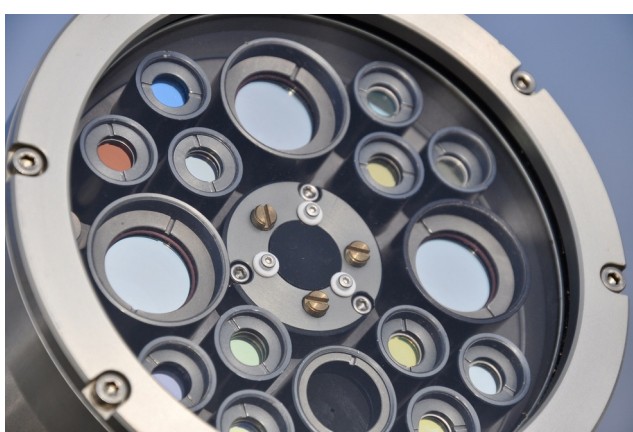

**Figure 1.** SSARA sensorhead with 12 direct channels (smaller diameter tubes), and 3 polarized channels (larger diameter at the top, left and right). The quadrant sun-tracker is in the center, below it is a finder for manual sun tracking.

## 2 Sun and sky scanning radiometer SSARA

### 2.1 Instrument characterization

SSARA is a multispectral sun photometer that has been designed and built at the Meteorological Institute Munich. The instru-
ment consists of three main components. These are the sensorhead, an alt–azimuthal mount, and a controller box containing a
Programmable Logic Controller (PLC). The latter is responsible for actuating the mount, operating the sensorhead with all its
life support, and digitizing the sensorhead's signals.

The radiometer's sensorhead (Fig. 1) houses baffles for 15 channels with a nominal field of view (FOV) of $1.2°$. The selection
of wavelengths for the channels is done by bandpass interference filters in front of the baffles. Their characteristics are given in
Table 1. All channels are installed parallel to each other with a quartz glass window, allowing for simultaneous measurement.
The pointing of the channels is parallel to within 20 arcmin.

For channels 1–12, measurements of the direct sun radiance are possible. Their filters are chosen to have similar character-
istics to those used in AERONET instruments. In addition to the bandpass interference filters, channels 13–15 are equipped
with linear polarizers. These are oriented at roughly $0°$, $-45°$ and $90°$ relative to the sensorhead's horizontal axis. Channels 3,
7, and 11, as well as the polarized channels, have a second amplifier stage installed. This allows for measurements of the sky
radiance, which is several orders of magnitude smaller than the direct sun radiances. These measurements are performed in the
solar principal and the almucantar plane. Furthermore, the sensorhead includes a four-quadrant sensor for tracking the sun.

The instrument can perform measurements at a maximum time resolution of about 1.6 s, which is used for the direct mea-
surements. Due to the design of the electronics, the amplifiers of the polarized channels have a higher time constant of 1 s
(compared to 0.25 s in the direct channels). For scans, we therefore wait 6 s to allow for the detector signal to settle, preventing
the measurements at different scanning angles from "blurring" into one another.





**Table 1.** SSARA channel configuration from 23 January 2017 onward. $\lambda_{\mathrm{ctr}}$ is the central wavelength of the filter, $\Delta\lambda$ is its full-width at half maximum. *gain* gives the amplification of the second amplifier stage, if installed for the corresponding channel.

| Nr. | $\lambda_{\mathrm{ctr}}$ [nm] | $\Delta\lambda$ [nm] | gain | remarks |
|---|---|---|---|---|
| 1 | 340.2 | 1.9 | | |
| 2 | 378.7 | 1.9 | | |
| 3 | 440.2 | 10.1 | 211.0 | |
| 4 | 499.8 | 9.8 | | |
| 5 | 614.8 | 3.6 | | |
| 6 | 675.7 | 9.8 | | |
| 7 | 780.8 | 5.8 | 210.5 | |
| 8 | 869.6 | 9.7 | | |
| 9 | 909.7 | 9.8 | | for water vapor abs. |
| 10 | 936.6 | 9.7 | | for water vapor abs. |
| 11 | 1020.4 | 9.7 | 1004.8 | damaged |
| 12 | 1639.7 | 25.3 | | InGaAs detector |
| 13 | 501.5 | 7.9 | 2.0 | polarized, 0° |
| 14 | 501.5 | 7.9 | 2.0 | polarized, -45° |
| 15 | 501.5 | 7.9 | 2.0 | polarized, 90° |

The sensorhead is mounted on a two-axis alt–azimuthal mount (Seefeldner et al., 2004). Its stepper motors have a resolution of 0.009° (32.4 arcsec). In order to apply proper corrections to the Rayleigh scattering background, the air pressure is recorded as well. The sensorhead is continuously heated to 40 °C to minimize drifts in sensor and filter characteristics.

5    Sunlight scattered from the glass window and possible dirt particles on it can create straylight, especially at larger scattering angles. To minimize this effect, a baffle has been designed and built in preparation of the A-LIFE campaign. It consists of a 24 cm long, black PVC cylinder with openings for the channels, leaving a 2 mm clearing to their FOV. This should inhibit direct sun light from hitting the front glass for scattering angles greater than 3.5°.

The scan patterns and wavelengths of SSARA are similar to those of the Cimel instruments used in AERONET, allowing for
10  comparison. However, in contrast to Cimel, it is able to measure all its channels simultaneously, because it does not use a filter wheel. Also, since it is not part of an operational network, it can be operated in any mode deemed appropriate. For instance, sky radiance scans can be performed at a higher rate, or even using new patterns for testing.



## 2.2 Calibration

### 2.2.1 Polarmetric calibration

The polarimetric and radiometric calibration of the sky radiance channels has been performed at Laboratoire d'Optique At-
mosphérique (LOA) in Lille, France. To produce linear polarized light a combination of an Ulbricht sphere and the so-called
POLBOX was used (Balois, 1998). Figure 2 depicts the calibration setup.

The POLBOX acts as a linear polarizer for the unpolarized light coming from the sphere. It consists of two glass plates
that can be tilted up to $65°$ relative to the optical axis. According to the Fresnel equations, the total attenuation exerted by a
glass plate differs for radiation polarized in the incident plane ($I_\parallel$) and perpendicular to it ($I_\perp$). Therefore, the degree of linear
polarization (DoLP) $\eta$ of the transmitted light is higher than that of the incident light. This degree of linear polarization is
hereby dependent on to the tilting angle of the glass plate $\alpha$.

$$\eta(\alpha, n) = \frac{I_\parallel - I_\perp}{I_\parallel + I_\perp} \tag{1}$$

$$= \frac{(1 - n^2)\left[\cos^2\alpha - \left(1 - \frac{1}{n^2}\sin^2\alpha\right)\right]}{(1 + n^2)\left[\cos^2\alpha + \left(1 - \frac{1}{n^2}\sin^2\alpha\right)\right]} \tag{2}$$

Since the angle of the plate can be determined with high accuracy, also the DoLP is known to a high precision. The entire
assembly can be rotated around its optical axis, therefore changing the polarization plane of the transmitted light. When using
two plates and tilting the second by the same angle $\alpha$, but in the opposite direction, a divergent ray of light hitting the first plate
at angle $\alpha + \delta\alpha$ will hit the second plate at an angle $\alpha - \delta\alpha$. This compensates for linear terms of error in the DoLP due to
divergent light. It can be shown that the DoLP after the second plate $\eta_{\text{tot}}$ is given by

$$\eta_{\text{tot}}(\alpha, n)$$

$$= \frac{2\eta(\alpha)}{1 + \eta^2(\alpha)} + \mathcal{O}\left(\delta\alpha^2\right) \tag{3}$$

$$\approx \frac{(1 - n^4)\left(\cos^4\alpha - \cos^4\alpha'\right)}{(1 + n^4)\left(\cos^4\alpha + \cos^4\alpha'\right) + 4n^2\cos^2\alpha\cos^2\alpha'} \tag{4}$$

Here, $\alpha$ is the angle between the incident light and the normal of the glass plate, $\alpha'$ is the same, but for the exiting light. $\alpha'$
can be calculated using Snellius law.

$$\sin\alpha = \sin\alpha' \cdot n \tag{5}$$

$$\Rightarrow \quad \cos\alpha' = \sqrt{1 - \frac{1}{n^2}\sin^2\alpha} \tag{6}$$

The refractive index of air is assumed to be 1. The plates are fabricated from *Schott SF-11* type glass. Its datasheet provides
coefficients for the Sellmeier equation (Eq. (7)), to calculate the refractive index $n$:

$$n(\lambda) = \sqrt{1 + \sum_i \frac{B_i\lambda^2}{\lambda^2 - C_i}}, \tag{7}$$


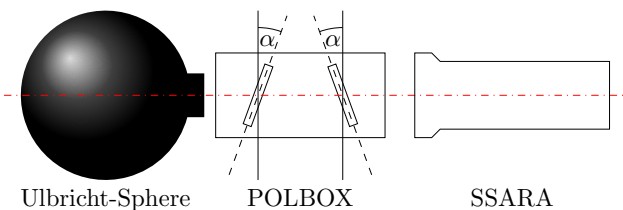

**Figure 2.** POLBOX calibration setup

with

$$B_1 = 1.73759695,$$

$$B_2 = 0.313747346,$$

$$B_3 = 1.898781010,$$

$$C_1 = 0.01318870700\,\mu\mathrm{m}^2,$$

$$C_2 = 0.0623068142\,\mu\mathrm{m}^2,$$

$$C_3 = 155.2362900\,\mu\mathrm{m}^2.$$

The POLBOX has a maximum tilt angle of $\alpha = 65°$. The resulting DoLP is roughly $58\%$ at the SSARA polarized wavelength of $501.5$ nm.

Polarized radiation can be described by what is known as the *Stokes vector* $\boldsymbol{S}$ (Chandrasekhar, 1950). It describes its total intensity, as well as its polarization state.

$$\boldsymbol{S} = \begin{pmatrix} I \\ Q \\ U \\ V \end{pmatrix} = \begin{pmatrix} E_x^2 + E_y^2 \\ E_x^2 - E_y^2 \\ 2E_x E_y \cos\delta \\ 2E_x E_y \sin\delta \end{pmatrix}, \tag{8}$$

where $E_x$ and $E_y$ is the strength of the electromagnetic radiation in the two transversal directions. $\delta$ is the phaseshift between these two components. $I$ describes the total intensity, $Q$ and $U$ the intensity of the linear polarized contribution, and $V$ that of circular polarization. As a result, the first component has to be larger or equal to the sum of the others. In atmospheric radiative transfer, the contribution of circular polarization is orders of magnitude smaller compared to linear polarization (e.g., de Haan et al., 1987; Emde et al., 2010), so it can be ignored here ($V \approx 0$). This leads to the definition of the degree of linear polarization (DoLP) $\eta$,

$$I \geq \sqrt{Q^2 + U^2}, \tag{9}$$

$$\Rightarrow \quad \eta = \frac{\sqrt{Q^2 + U^2}}{I}. \tag{10}$$

In the *Stokes–Müller* formalism, interactions with optical components or the atmosphere are described by left multiplication of the Stokes vector of the incoming radiation $\boldsymbol{S_{\mathrm{in}}}$ with the appropriate real $4 \times 4$ Müller matrices $\mathbf{M_1}$ to $\mathrm{M_n}$,

$$\boldsymbol{S_{\mathrm{out}}} = \mathbf{M_n} \cdots \mathbf{M_1} \cdot \boldsymbol{S_{\mathrm{in}}}. \tag{11}$$





In this context, a linear polarizer can be described as a linear diattenuator, meaning its attenuation differs for the two directions of polarization. The Müller matrix $\mathbf{LD}$ for a linear diattenuator rotated by an arbitrary angle $\vartheta$ is given in Bass et al. (2010) as

$$\mathbf{LD}\left(\vartheta\right) =$$
$$\frac{1}{2}\begin{pmatrix} a & b\cos\left(2\vartheta\right) & b\sin\left(2\vartheta\right) & 0 \\ b\cos\left(2\vartheta\right) & a\cos^2\left(2\vartheta\right)+c\sin^2\left(2\vartheta\right) & \left(a-c\right)\cos\left(2\vartheta\right)\sin\left(2\vartheta\right) & 0 \\ b\sin\left(2\vartheta\right) & \left(a-c\right)\cos\left(2\vartheta\right)\sin\left(2\vartheta\right) & a\sin^2\left(2\vartheta\right)+c\cos^2\left(2\vartheta\right) & 0 \\ 0 & 0 & 0 & c \end{pmatrix}, \tag{12}$$

with $a = k_0+k_1$, $b = k_0-k_1$, and $c = 2\sqrt{k_0 k_1}$. $k_0$ and $k_1$ are the transmission values for the filter in the direction parallel and perpendicular to its orientation, respectively. $\vartheta$ is the angle between the polarization direction of the incoming radiation and the filter. Since a photodiode can only measure the total intensity of the light (first component of Stokes vector), the measurement operator $\langle M|$ projects only the first row of the matrix. Mathematically, it can be described as a transposed vector $(1,0,0,0)$

$$I = \langle M|\mathbf{LD}|\boldsymbol{S}\rangle \tag{13}$$
$$= \frac{1}{2}\left[a\cdot I_0 + b\cdot\cos\left(2\cdot\Delta\vartheta\right)\cdot Q_0 + b\cdot\sin\left(2\cdot\Delta\vartheta\right)\cdot U_0\right] \tag{14}$$

The light entering the instrument behind the POLBOX is taken to be polarized only in the positive $Q$ direction. This means the Stokes vector is given by $(I_0, \eta_{\mathrm{tot}} I_0, 0, 0)^{\mathrm{T}}$, with $\eta_{\mathrm{tot}}$ again being the degree of linear polarization produced by the POLBOX. Also, the sensor has a certain radiometric response $C$, so the measurement vector becomes $\langle M| = (C, 0, 0, 0)$.

$$S\left(\vartheta\right) = \frac{C}{2}\left[a\cdot I_0 + b\cdot\cos\left(2\left(\vartheta-\vartheta_0\right)\right)\cdot\eta\cdot I_0\right] \tag{15}$$
$$= \frac{1}{2}\left[a'\cdot I_0 + b'\cdot\cos\left(2\left(\vartheta-\vartheta_0\right)\right)\cdot\eta\cdot I_0\right] \tag{16}$$
$$= \frac{1}{2}\left[A' + \eta\cdot B'\cdot\cos\left(2\left(\vartheta-\vartheta_0\right)\right)\right] \tag{17}$$

It can be seen that the polarimetric (described by $a$ and $b$) and radiometric response ($C$) of the instrument/filter combination cannot be determined separately. Therefore, we introduce $a' = C\cdot a$ and $b' = C\cdot b$. Also, since the total intensity of the incoming light is unknown, so we define $A' = a'\cdot I_0$ and $B' = b'\cdot I_0$. Measuring the signal $S$ at varying rotation angles $\vartheta$ of the POLBOX, the parameters $A'$, $B'$ and $\vartheta_0$ can be obtained by performing a Levenberg–Marquardt (LM) fit using Eq. (17) as a model. $k_0$ and $k_1$ cannot be determined independently, but it is possible to derive the diattenuation $D$ as

$$D = \frac{\left(k_0-k_1\right)}{\left(k_0+k_1\right)} = \frac{b}{a} = \frac{b'}{a'} = \frac{B'}{A'}. \tag{18}$$

It is independent of the intensity of the incoming radiation $I_0$. The LM-fit also gives estimations for the uncertainties in $A'$, $B'$, and $\vartheta_0$. For determining the response $a'$, we use LOA's *SphereX*, a radiometrically calibrated Ulbricht sphere. As it provides unpolarized light with known intensity, the measured signal is given by

$$S = \frac{C}{2}a\cdot I_0 = \frac{a'}{2}I_0 \tag{19}$$
$$\Rightarrow\quad a' = \frac{2\cdot S}{I_0}. \tag{20}$$





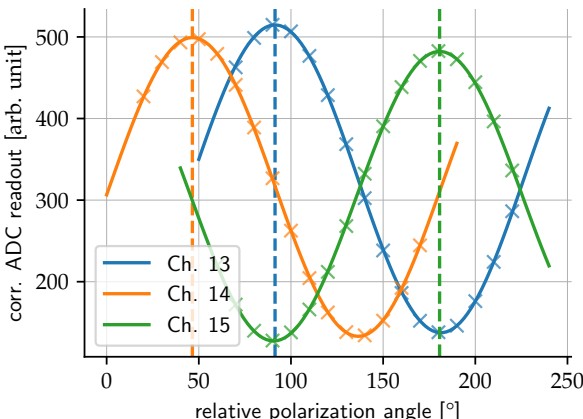

**Figure 3.** Fit of Eq. (17) to intensity measurements of the three polarized SSARA channels at varying POLBOX orientations. The dashed horizontal lines correspond to the angles of maximum transmission $\vartheta_0$. Amplitude and vertical offset are related to the radiometric and polarimetric response.

**Table 2.** Calibration results for measurements on 2 February 2017. The uncertainties are determined from the fit.

| Channel | $\vartheta_0$ [°] | $D$ [-] | $a'$ [1/mW m$^{-2}$] |
|---|---|---|---|
| 3 | - | 0 | 412 |
| 7 | - | 0 | 331 |
| 11 | - | 0 | 362 |
| 13 | $91.36 \pm 0.06$ | $0.984 \pm 0.002$ | 8164 |
| 14 | $46.51 \pm 0.05$ | $0.985 \pm 0.002$ | 7979 |
| 15 | $180.62 \pm 0.07$ | $0.990 \pm 0.002$ | 7717 |

Channels 3, 7, and 11, are unpolarized, so, per definition, $k_0 = k_1 = 1$, and therefore $D = 0$.

For the SSARA calibration on 2 February 2017, the fit of Eq. (17) to the measurements can be seen in Fig. 3. The determined
5 values and their uncertainties are shown in Table 2. It should be noted that the sensorhead was placed on its right side, therefore adding roughly 90° to the filter orientation.

What remains after this calibration is the collective rotation of all channels in the sensorhead, which also includes rotations stemming from the mount. When only the degree of linear polarization is of interest, this is not relevant. However, this global rotation has to be known to determine the polarization angle, which influences how the polarized radiation is devided between the $Q$ and $U$ component. As outlined in Li et al. (2014), this could be done by using known features of the Rayleigh background (e.g. $U = 0$ in principal plane).

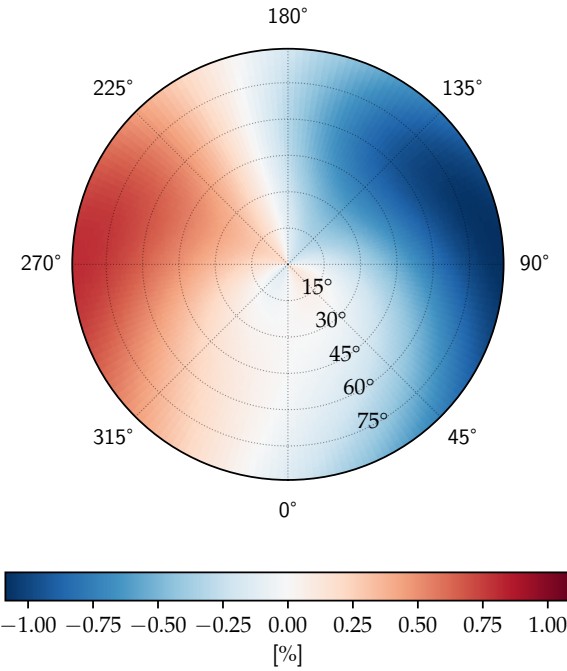

**Figure 4.** Relative difference in measured total radiance at 500 nm due to incorrect rotation and imperfect polarizer for a synthetic scene. The sun is at a zenith angle of 30° and azimuth 0°.

To determine the potential error arising from neglecting the imperfections of the filters and their orientation, a polarized

5 radiance all-sky panorama has been simulated for 500 nm using the *MYSTIC* 3D Monte–Carlo solver (Mayer, 2009; Emde et al., 2010) of the *libRadtran* package (Mayer and Kylling, 2005; Emde et al., 2016). To get the maximum error corresponding to the highest possible degree of linear polarization, a pure Rayleigh atmosphere was used as model input, without aerosol or clouds. Scattering processes by these would "destroy" polarization. The ground is non-reflective for the same reason and the sun at a zenith angle of 30°. The simulation is used to generate synthetic measurements in the three polarized SSARA channels,

10 taking into account the filter characteristics from Table 2. From these, the Stokes vector is reconstructed, once assuming perfect polarizers ($D = 1$) at exact angles (90°, 45°, and 180°), and again with the actual filter characteristics in Table 2. Their relative difference in the total radiance and the degree of linear polarization is displayed in Figs. 4 and 5, respectively. The relative error in total radiance varies by between -1.1 % and +0.8 %, the relative error in DoLP from -2.4 % and +1.5 % (relative, not in absolute value). Due to the relative rotation of the polarizers, the pattern is not symmetrical.

### 2.2.2 Mount calibration

SSARA should be set up perfectly perpendicular to the local tangential plane, facing exactly south. However, often this is possible only to within a few degrees. Also, SSARA is designed to be portable, so the setup procedure has to be performed



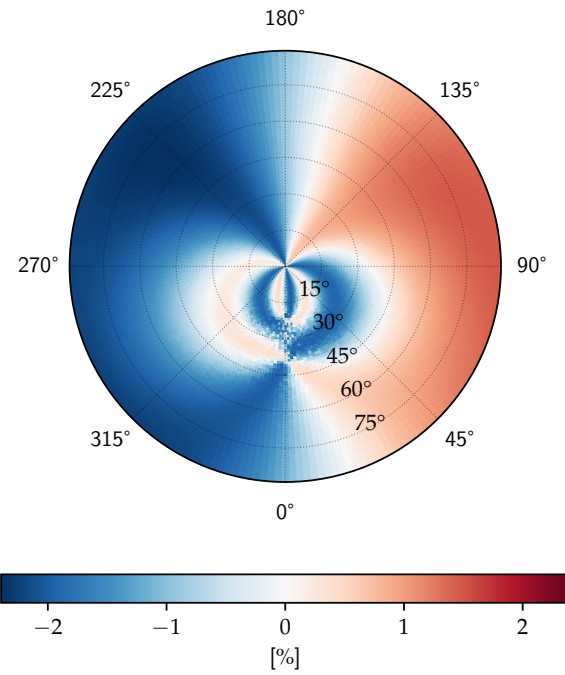

**Figure 5.** Same as Fig. 4, but for the relative difference in the degree of linear polarization

regularly. Therefore, it is useful to be able to quickly install the instrument in roughly the right orientation and determine the

5  exact alignment by correlating the positions of the mount motors with the known sun position for times with accurate sun
tracking.

To determine the actual orientation of the mount from several known sun positions, we cannot directly fit the Euler angles
using conventional real $3 \times 3$ rotation matrices, as this approach suffers from what is known as *gimbal lock*. This results from
singularities in spherical coordinate systems, caused by directional "flips", for instance when crossing the zenith. Conventional

10  minimization methods are not applicable in such highly non-linear cases. However, the fit can be performed using quaternions,
as rotations here are always smooth and free of singularities. The mathematical fundamentals of quaternions are given in
Appendix A. To perform the mount calibration, several coordinate systems are defined that can be transformed into one another
by rotation. Translation is ignored, as the earth-sun-distance is much larger than the replacements in the instrument and mount.
The coordinate systems used are similar to those defined in Riesing et al. (2018). Figure 6 sketches the coordinate systems
used for SSARA:

– *East-North-Up (ENU)*: local horizon coordinate system on the tangential plane containing the observation position.
Elevation and azimuth of the sun ($\vartheta_s$, $\phi_s$) can be calculated for this system. $x$-axis points towards east, $y$-axis towards
north, and $z$-axis towards zenith.




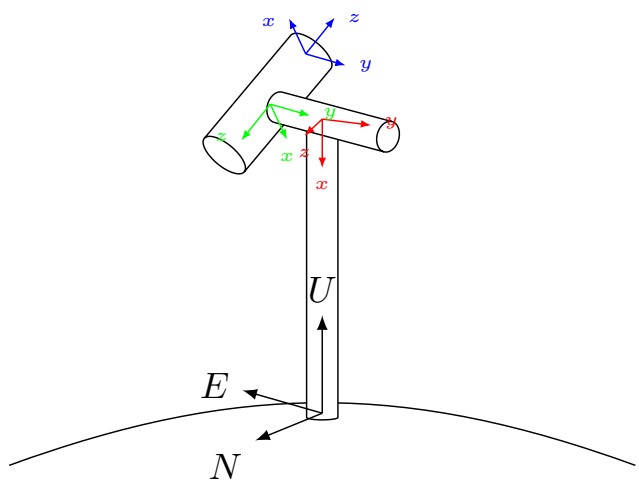

**Figure 6.** Sketch of the SSARA instrument and the coordinate systems used for the mount calibration; ENU (black), unrotated mount system MNT (red), gimbaled mount system GMB (green), and sensorhead system SH (blue).

- *Mount (MNT)*: $y$-axis along that of the elevation motor, $x$-axis is along the rotation axis of the azimuth motor, with the elevation motor centered ($\phi_0 = 0$). $z$-axis is the cross product of $x$ and $y$-axis to form a right-handed system.

- *Gimbaled system (GMB)*: the mount system rotated around the motor axes by the elevation $\vartheta$ and azimuth $\phi$. These angles consist of the zero-offset of the motor axes ($\vartheta_0$ and $\phi_0$), and the rotation of the motors ($\Delta\vartheta$ and $\Delta\phi$). By choice of the MNT system, $\phi_0$ is defined as zero. Additionally, a non-perpendicularity between the motor axes $\delta$ is considered.

- *Sensorhead (SH)*: $z$-axis points along the optical axis of the sensorhead, $x$-axis points towards the top of the instrument, $y$-axis towards the right, forming a right-handed system.

In an ENU spherical coordinate system, the azimuth $\phi$ is zero in the north and increases towards the east, as one would expect. The polar angle is zero in the nadir and increases towards the zenith. Rotations between the coordinate systems are described by quaternions, where $^{\mathrm{B}}\boldsymbol{q}_{\mathrm{A}}$ is a quaternion rotating coordinate system $A$ to $B$.

For direct measurements with the quadrant sensor uniformly lit, the sun and viewing vector in the ENU system are assumed to be equal (to within the accuracy of the suntracker). The sun position in the ENU system is determined with the *pyEphem* Python package. It can calculate planetary positions to a precision satisfactory for our purpose using the VSOP87 model (Bretagnon and Francou, 1988). To obtain the viewing vector $\boldsymbol{r}_v$ of the instrument, the unit vector in $z$-direction $\boldsymbol{e}_z$ in the *SH* system has to be transformed as follows

$$\boldsymbol{r}_v = {}^{\mathrm{ENU}}\boldsymbol{q}_{\mathrm{SH}}\boldsymbol{r}_{\mathrm{SH}} = {}^{\mathrm{ENU}}\boldsymbol{q}_{\mathrm{SH}}\boldsymbol{e}_z\,. \tag{21}$$





The optimal rotation quaternion can be found by minimizing the distance between viewing vectors and sun vector $\boldsymbol{r}_v$,

$$S = \frac{1}{N} \sum_{i=0}^{N} \|\boldsymbol{r}_s - \boldsymbol{r}_v\| = \frac{1}{N} \sum_{i=0}^{N} \left\|\boldsymbol{r}_s - {}^{\mathrm{ENU}}\boldsymbol{q}_{\mathrm{SH}} \boldsymbol{e}_z\right\| . \tag{22}$$

However, ${}^{\mathrm{ENU}}\boldsymbol{q}_{\mathrm{SH}}$ is composed of several rotations:

$${}^{\mathrm{ENU}}\boldsymbol{q}_{\mathrm{SH}} = {}^{\mathrm{ENU}}\boldsymbol{q}_{\mathrm{MNT}}{}^{\mathrm{MNT}}\boldsymbol{q}_{\mathrm{GMB}}{}^{\mathrm{GMB}}\boldsymbol{q}_{\mathrm{SH}} \tag{23}$$

${}^{\mathrm{GMB}}\boldsymbol{q}_{\mathrm{SH}}$ is defined as a 180° rotation around the local $y$-axis to obtain the sensorhead coordinate system. The active com-

ponent of the mount acts on ${}^{\mathrm{MNT}}\boldsymbol{q}_{\mathrm{GMB}}$. It contains the rotation angles of the azimuth and elevation motors ($\Delta\phi$ and $\Delta\vartheta$), as well as the zero-point offset angles of the motors ($\vartheta_0$ and $\phi_0$). $\phi_0$ is zero due to our definition of the MNT system (it is effectively absorbed into ${}^{\mathrm{ENU}}\boldsymbol{q}_{\mathrm{MNT}}$), but $\vartheta_0$ has to be determined. Both offset angles are constant over time and do not change for instrument realignment. Furthermore, the non-perpendicularity $\delta$ between the two motors is considered.

$${}^{\mathrm{MNT}}\boldsymbol{q}_{\mathrm{GMB}}$$

$$=\boldsymbol{q}\left(\phi, \boldsymbol{e}_x\right)\left[\boldsymbol{q}\left(\delta, \boldsymbol{e}_z\right)\boldsymbol{q}\left(\vartheta, \boldsymbol{e}_y\right)\boldsymbol{q}\left(-\delta, \boldsymbol{e}_z\right)\right] \tag{24}$$

$$=\boldsymbol{q}\left(\phi_0 + \Delta\phi, \boldsymbol{e}_x\right)$$
$$\left[\boldsymbol{q}\left(\delta, \boldsymbol{e}_z\right)\boldsymbol{q}\left(\vartheta_0 + \Delta\vartheta, \boldsymbol{e}_y\right)\boldsymbol{q}\left(-\delta, \boldsymbol{e}_z\right)\right] \tag{25}$$

$$=\boldsymbol{q}\left(\Delta\phi, \boldsymbol{e}_x\right)$$
$$\left[\boldsymbol{q}\left(\delta, \boldsymbol{e}_z\right)\boldsymbol{q}\left(\vartheta_0, \boldsymbol{e}_y\right)\boldsymbol{q}\left(\Delta\vartheta, \boldsymbol{e}_y\right)\boldsymbol{q}\left(-\delta, \boldsymbol{e}_z\right)\right] \tag{26}$$

${}^{\mathrm{ENU}}\boldsymbol{q}_{\mathrm{MNT}}$ is unknown and contains the tilt and rotation of the mount. It changes every time the instrument is moved, involving a new calibration. The minimization now has six variables (four components of ${}^{\mathrm{ENU}}\boldsymbol{q}_{\mathrm{MNT}}$, $\delta$, and $\vartheta_0$), and one constraint (${}^{\mathrm{ENU}}\boldsymbol{q}_{\mathrm{MNT}}$ has to be normed). This can be achieved using the *Sequential Least SQares Programming* (SLSQP) algorithm (Kraft, 1988).

    For the A-LIFE data, the fitting determines a non-perpendicularity of the motors $\delta$ of 0.95° and an elevation offset $\vartheta_0$ of

-6.46°. The rotation quaternion ${}^{\mathrm{ENU}}\boldsymbol{q}_{\mathrm{MNT}}$ is reconstructed to (0.704, -0.044, -0.707, 0.043). While the non-perpendicularity and the elevation offset are constant over time, the rotation quaternion will change every time the instrument is moved.

    Figure 7 shows the remaining deviation between the fitted instrument pointing and the actual sun position for all measurements in the A-LIFE campaign. The calibration is accurate to within 32 arcmin, corresponding to the apparent solar radius. The remaining inaccuracies are most likely due to the limited precision of the quadrant-sensor and the way the instrument is

tracking the sun. The sensor has to pick up on brightness-differences over the sun. Also, high aerosol loads, cirrus, or thin water clouds blur out the sun disc, resulting in an equally lit quadrant-sensor further away from the sun's center. If the clouds are "streaky", this effect can occur in a certain direction. To avoid oscillation of the sensorhead the correction of pointing is damped. As a result, the instrument will most likely point to the lower left of the sun disc in the morning, and the upper left in the evening. Other disruptions might occur by the instrument having to "search" the sun after every scan. In the future, this effect should be minimized by used online-fitting of the mount skewness. Furthermore, the change of the apparent solar position due to atmospheric refraction has been ignored.




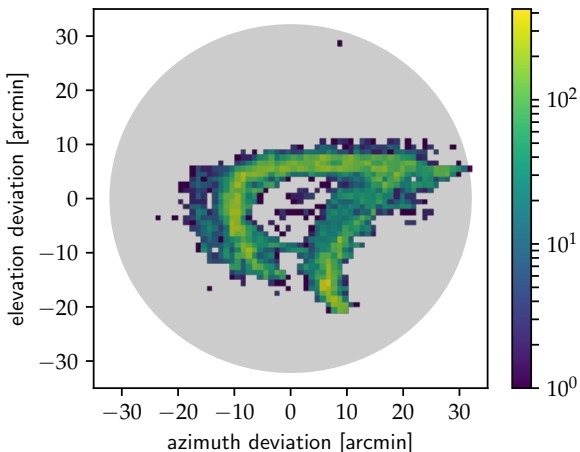

**Figure 7.** Residual between calibrated and calculated sun position. Average apparent size of sun disk (32 arcmin) as reference (grey).

### 2.2.3 Langley calibration

*Langley extrapolation* is a method to enable sun-photometers to retrieve the total optical depth of the atmosphere, without the need for a radiometric calibration of the instrument in a laboratory (Forgan, 1994). The basis for the extrapolation is the *Bouguer–Lambert–Beer* law and its logarithmic representation:

$$I = I_0 \cdot \exp(-m\tau) \tag{27}$$

$$\ln(I) = \ln(I_0) - m\tau \tag{28}$$

where $I$ and $I_0$ are the measured and extraterrestrial irradiance, respectively. $\tau$ is the optical depth, and $m$ the airmass factor. The latter describes the increase in the direct optical pathlength — and therefore the optical depth — from the sun to the detector. In the simplest geometric approach, $m = \cos^{-1}(\Theta)$, with the solar zenith angle $\Theta$. A more elaborate airmass model taking into account atmospheric refraction and the curvature of the earth can be found in Kasten and Young (1989). Additionally, the extraterrestrial irrandiance $I_0$ has to be corrected for the seasonal variablity in sun-earth distance (Spencer, 1971).

Taking measurements at varying values of the airmass factor, and assuming the optical depth to be constant over time, the logarithm of the irradiance in Eq. (28) can be fitted as a linear function of $m$ with slope $\tau$. Extrapolating the linear fit to $m = 0$ yields $\ln(I_0)$. This value can then be used for reconstructing $\tau$ from measurements of $I$. Since only the ratio of the irradiances $I$ and $I_0$ is used, they can be replaced by any detector signal $S$ that is linear in $I$.

$$\tau = \frac{1}{m}\ln\left(\frac{I}{I_0}\right) = \frac{1}{m}\ln\left(\frac{S}{S_0}\right), \tag{29}$$

with $S = C \cdot I$ and $\tau = \tau_R + \tau_M + \tau_A$.





**Table 3.** Boundaries and initial values for the aerosol parameters used in the retrieval. For effective variance ($v_{\mathrm{eff}}$), imaginary part of the refractive index ($m_{\mathrm{i}}$), and fraction of spherical particles ($f_{\mathrm{sph}}$) no bouds are given, as these quantities are fixed.

| Parameter | Fine mode | | | Coarse mode | | |
| --- | --- | --- | --- | --- | --- | --- |
| | min. | max. | init | min | max | init |
| $r_{\mathrm{eff}}$ [μm] | 0.05 | 0.5 | 0.1 | 0.5 | 3.0 | 1.0 |
| $v_{\mathrm{eff}}$ [-] | - | - | 0.62 | - | - | 0.62 |
| $m_{\mathrm{r}}$ [-] | 1.35 | 1.65 | 1.5 | 1.35 | 1.65 | 1.5 |
| $m_{\mathrm{i}}$ [-] | - | - | 0.01 | - | - | 0.01 |
| $\tau_{550}$ [-] | 0.01 | 1.0 | 0.1 | 0.01 | 1.0 | 0.05 |
| $f_{\mathrm{sph}}$ [-] | - | - | 1.0 | - | - | 0.1 |

This $\tau$ is the combined value of Rayleigh ($\tau_R$), trace gas ($\tau_M$), aerosol ($\tau_A$), and possibly cloud ($\tau_C$) optical depths. The

contribution from Rayleigh was determined according to Bodhaine et al. (1999), scaled with the measured air pressure. At around 500 nm, $O_3$ and $NO_2$ are the main contributors to the trace gas optical depth $\tau_M$. Their profiles were taken from Anderson et al. (1986), and the corresponding absorption cross-sections from Bogumil et al. (2003). Assuming that no clouds are present, subtracting these components from the total optical depth leaves only the contribution from aerosol.

SSARA is usually calibrated once a year, either around March/April or around October/November at UFS Schneefernerhaus

(2650 m) on Mount Zugspitze. Firstly, at a height of over 2600 m, the contamination by boundary layer aerosols is minimal. Also, early/late in the year, convective processes over the measurement site are not prominent. Therefore, temporal homogenity of $\tau$ is found more frequently during that time. The calibration used for the data presented in this paper was done in November 2016.

## 3 Retrieval of aerosol properties from SSARA observations

### 3.1 Retrieval algorithm

Our algorithm (Grob et al., 2019) minimizes the difference between observed polarized sky radiances and corresponding forward model simulations by varying aerosol properties. These retrieved aerosol parameters are effective radius $r_{\mathrm{eff}}$, real part of the refractive index $m_{\mathrm{r}}$, and optical depth (AOD) at 550 nm $\tau_{550}$ for two aerosol modes with a log-normal particle size distribution. Each of these quantities are retrieved separately for both modes. Table 3 shows all the initial values and retrieval

limits for all parameters of the aerosol model. If no boundaries are given, the parameter is not varied but fixed to its initial value. These are the effective variance of the particle size distribution $v_{\mathrm{eff}}$, the imaginary part of the refractive index $m_{\mathrm{i}}$, and the fraction of spherical particles $f_{\mathrm{sph}}$. The retrieval has previously been validated with synthetic observations of a variety of clear sky and cloudy situations with varying aerosols.





However, for this study several changes have been made compared to Grob et al. (2019) to better adapt the retrieval algorithm to measurements. Firstly, we assume a mixture of spherical and non-spherical particles for the coarse mode. This is more realistic for many aerosols (e. g. Dubovik et al., 2006, and references therein). The optical properties of this mixture is calculated by linear mixing of the tabulated optical properties for spheres and spheroids from Dubovik et al. (2006). They describe spheroids as a mixture of particles with aspect ratios ranging from 0.3 (elongated) to 3.0 (flattened). The fine mode is still assumed to contain only spherical particles. A ground albedo of 0.15 has been estimated from MODIS observations and is used for all wavelengths.

Additionally, the cloud-screening has been be revised. Due to the higher level of noise in the measuements, the original method classified too many measurements as cloudy. Furthermore, SSARA also provides unpolarized radiance measurments at 440 nm and 780 nm usable for cloud detection. In the new version, a set of 500 simulations of the given scan geometry is performed with aerosol parameters randomly sampled from the ranges given in Table 3. For simplicity and computational speed, only a single aerosol mode is used in these forward simulations. For every wavelength, the measured total radiance and its derivative with respect to the scattering angle are compared to these simulations. If the measured quantities are not within the 95th percentile of the simulated values, the measurement at this angle is flagged as cloudy. The same is done for the DoLP at 500 nm. This gives four separate cloud masks, three from unpolarized radiances at 440 nm, 500 nm and 780 nm, and one from the DoLP at 500 nm. If more than two of them indicate a cloud at a certain scan angle, this datapoint is removed from the scan for the subsequent retrieval. This multi-stage approach makes the method robust against noise, but still strict enough to reliably remove observations of clouds.

Finally, the measurement scans performed with SSARA during the A-LIFE campaign are not taken at equidistant scattering angles. Similar to scans performed by instrument in the AERONET framework, the measurements are denser around the sun. This results in this area being overrepresented and therefore overweighted in the minimization procedure. However, much of the additional information provided by polarization is contained in measurements at larger scattering angles. To account for this, all measurements are weighted by the inverse of their angular density

$$w_i = \frac{1}{2} \left( \vartheta_{i+1} - \vartheta_{i-1} \right),$$ (30)

where $w_i$ is the weight of the $i$th measurement point, and $\vartheta_i$ the corresponding scattering angle.

## 3.2 Case studies

The following measurements have been performed during the A-LIFE field campaign. SSARA was installed on top of a building of the University of Cyprus at Limassol (N 34.674°, E 33.040°). The AERONET station CUT-TEPAK is installed about 300 m to the east. The Leipzig Aerosol and Cloud Remote Observations System (LACROS, Bühl et al. (2013)), including a Polly$^{XT}$ lidar system (Engelmann et al., 2016), was located 400 m to the north east.

Between 6 and 28 April, SSARA continously performed direct sun observations. These have been interleaved with sky radiance scans in the almucantar and principal plane at pre-selected solar zenith angles. Almucantar plane scans have been carried out at every 5° of solar zenith angle between 35° and 80°, principal plane at 10° intervals between 30° and 80°.



The data of channel 11 (1020 nm) were excluded from the analysis as it intermittently provided faulty values during the measurement campaign.

For testing our retrieval data from 17 and 20 April were selected for more in depth case studies. To evalutate of the retrieval performance the same criteria were used as in the numerical studies. These were taken from Mishchenko et al. (2004) and allow for a maximum deviation of 0.04 or 10 % deviation in AOD, 0.1 μm or 10 % in effective radius, and 0.02 in the refractive index. Since the true value is unknown, the results were compared with the AOD retrieved from direct sun observations and the level 1.5 data of the AERONET version 3 inversion. Level 1.5 data were used, since level 2.0 did not include refractive index

values for the chosen dates. It should be noted that the AERONET inversion uses the same refractive index for both modes.

Since the plots showing the results the same for both days, they will be described here first. Figures 9 and 12 show the aerosol optical depth at 500 nm for these two days. Orange and blue crosses mark values retrieved by the inversion from principal plane and almucantar scans, respectively. The residual in the minimization is shown as an indicator of the performance of the retrieval for a given measurement. The values obtained from direct sun observations are displayed as reference, with green

dots representing AERONET L2 data and the red ones SSARA measurements. Figures 10 and 13 show all retrieved aerosol parameters for fine and coarse mode, separately. Again, blue corresponds to values obtained from principal plane, orange from almucantar scans. The AERONET points are the results of the AERONET inversion for hybrid (red) and almucantar scans (green). Since AERONET uses a common refractive index for fine and coarse mode, this value is shown for both modes (subplots (e) and (f)). It should amount to a weighted mean of the values we retrieved for the two modes, and therefore lie

somewhere between those. To facilitate the comparison of the retrieval results with direct sun measurements and AERONET values, the optical depth is evaluated at 500 nm in the following case studies.

### 3.2.1   Cloudy day (17 April 2017)

17 April has been chosen to illustrate the retrieval behaviour during cloudy phases. Around sunrise and between roughly 11:00 UTC and 14:15 UTC, convective clouds have been present at the measurement site. This can also be deduced from

25   the gap in AERONET direct sun AOD data. Cirrus clouds already appeared around 10:30 UTC, and persisted almost until 16:00 UTC. Figure 8 shows four snapshots of the cloud situation during that day. The pictures have been taken with a camera installed coaxially with the SSARA sensorhead.

In the early morning (until around 04:30 UTC Fig. 9) an elevated AOD is retrieved. This coincides with the presence of convective clouds also visible in the top right panel of Fig. 8. As shown in sensitivity studies, these might lead to an

30   overestimation of the AOD. However, it could indicate that additionally the AOD is increased, for example due to hygroscopic growth of aerosol particles in humid air. The same can be observed in Fig. 9 for the convective period in the afternoon between 11:00 UTC and 13:00 UTC. Here it should be noted that for the corresponding scans, the residual is sometimes slightly higher, indicating a less reliable retrieval result. This is shown by the black tickmarks in Figs. 9 and 10. Most of the time, the residual is below 0.1, but spikes up to 0.4. Until around 07:00 UTC, the retrieved total AOD is consistent with the values obtained from direct sun measurements. Starting around this time, the AOD is overestimated by up to 0.1 during clear sky periods. Small gaps in the AERONET direct measurements indicate the presence of clouds or high variability in the aerosol. Again, some



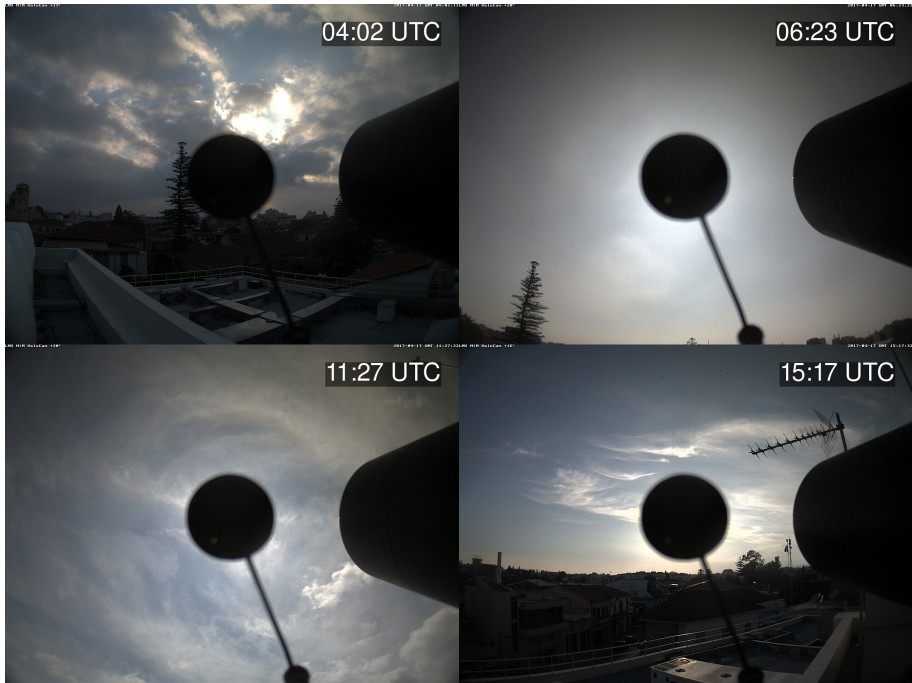

**Figure 8.** Sky camera images for 17 April 2017. The convective clouds in the early morning and afternoon are visible. The persisting cirrus clouds towards the evening can be seen.

deviation in the retrieval (generally overestimation) is to be expected here. Towards the evening, the optical depth seems to be underestimated. Note that perfect agreement between the values retrieved from sky radiance observation and from direct sun

observations cannot be expected. The reason for this might be inhomogeneity in the aerosol, either in space (maritime towards ocean, anthropogenic aerosols towards city/industry), or in time, as one scan can take up to 15 min. Other explanations could be measurement errors or systematic effects of the retrieval. This can also explain the differences between the results of almucantar and principal plane scans.

In Fig. 10a and Fig. 10b, the AOD is separated into fine and coarse mode. Over the entire day, the aerosol optical depth is

dominated by the fine mode. This compares well to the AERONET inversion datapoints. The contribution of the coarse mode is larger compared to AERONET. It should be noted here that — in contrast to the AERONET inversion — we do not use the total AOD from direct sun observations as a constraint for our minimization. This is not feasible for a method designed to be employed in cloudy situations, where such measurements might not be available.

The retrieved effective radius of the fine mode (Fig. 10c) is mostly consistent over the entire day, including the cloudy period

in the afternoon. This insensitivity of the effective radius to the presence of clouds was also observed in the numerical studies. However, the increased values in the morning and evening should be noted. This seems to be a systematic pattern, the reason for which is still unknown. When compared to AERONET our fine mode effective radii are somewhat smaller, but within the 0.1 μm limit. The same is true for the coarse mode (Fig. 10d). Here, the AERONET inversion suggests the presence of





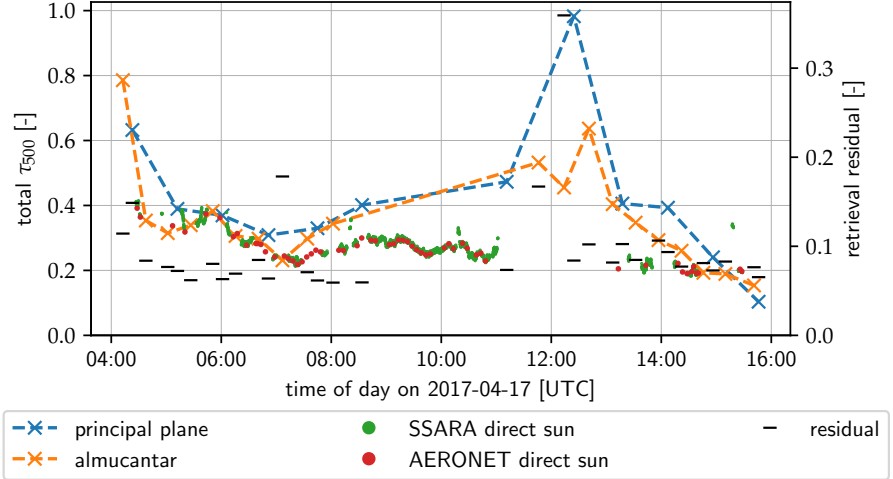

**Figure 9.** Total AOD at 500 nm for 17 April 2017. Crosses indicate values retrieved from SSARA almucantar (orange) and principal plane (blue) scans. Green and red dots are from direct sun observations with SSARA and AERONET, respectively. The thin black markers represent the residual of the retrieved solution.

large particles with an effective radius of around 2 μm between 07:00 UTC and 10:00 UTC. The values we obtain are smaller.

Although previous sensitivity studies have shown that our retrieval has the tendency to underestimate the size of large coarse mode particles, independent measurements would be needed to further investigate the discrepancy.

The retrieved real part of the refractive index changes rapidly for fine mode particles (Fig. 10e). High values can be observed in the aforementioned times with clouds present. This behaviour is again consistent with the results of the numerical studies, where clouds induce an overestimation of the index of refraction. The results for the coarse mode (Fig. 10f), are smoother in

general. The retrieved value mostly stays close to the prior of 1.5, which might be caused by a low sensitivity to this parameter. The refractive index derived from AERONET ranges from 1.33 to 1.48. At around 07:00 UTC there is an obvious discrepancy between values obtained from hybrid and almucantar scans. The values below 1.35 between 08:30 UTC and 10:00 UTC seem unrealistic, as all expected aerosol types have a higher refractive index.

### 3.2.2 Clear-sky day with arriving Saharan dust layer (20 April 2017)

20 April was a clear-sky day. Starting in the late morning (07:00 UTC, 10:00 LT), the AOD increased. This can be attributed to the arrival of a Saharan dust outbreak over Cyprus from the west. Figure 11 shows the attenuated backscatter at 1064 nm of the Polly$^{XT}$ lidar. An aerosol layer is visible between roughly 2 km and 5 km beginning with thin filaments at around 04:00 UTC, and increasing in thickness towards noon. Polly$^{XT}$ also provides measurements of the particle linear depolarization ratio (PLDR) at 532 nm that can be used to discriminate between types of aerosol (Baars et al., 2016). In this layer, PLDR values around 25 % are observed and clearly identify the aerosol as desert dust (Müller et al., 2003; Freudenthaler et al., 2009).



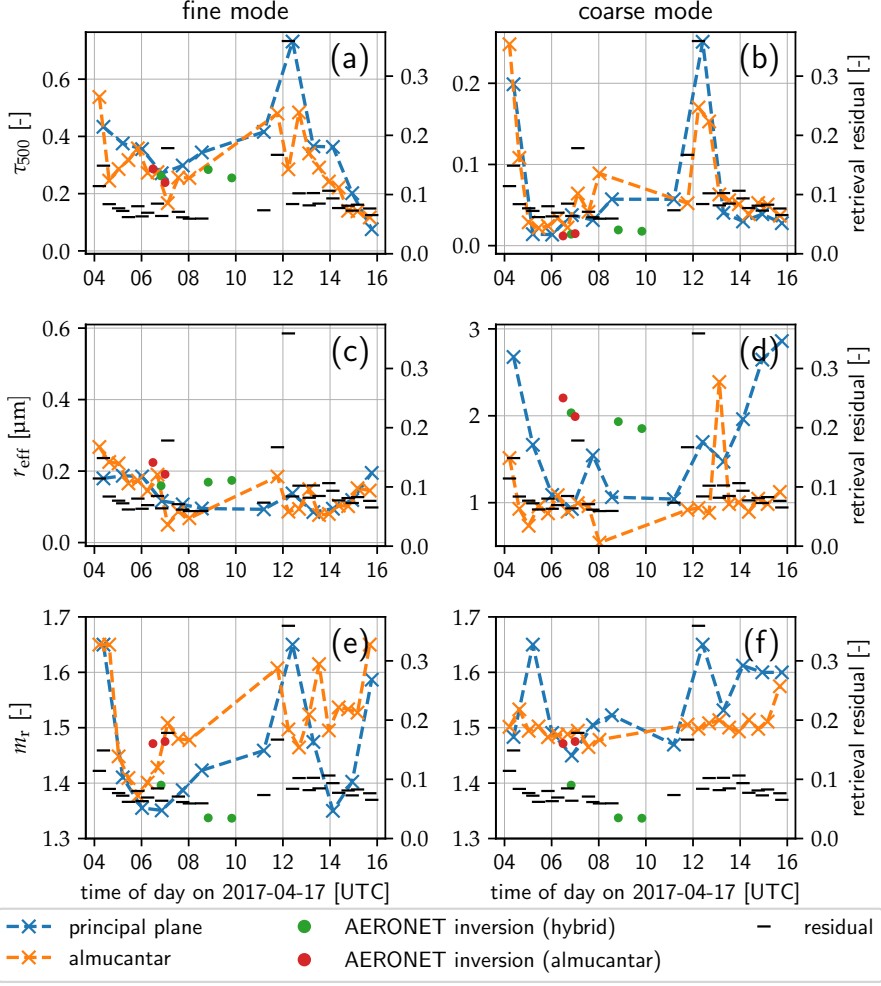

**Figure 10.** Fine and coarse mode trends of aerosol optical depth ($\tau_{500}$), mode effective radius ($r_{\text{eff}}$), and real part of refractive index ($m_r$) for 17 April 2017. Values obtained from principal plane scans are marked by blue crosses, those from almucantar scans are orange. The residual of the retrieval is shown by black markers. AERONET version 3 level 1.5 inversion results are shown as green and red dots, corresponding to retrieval of almucantar and hybrid scans, respectively. The refractive index is assumed to be equal for both modes in the AERONET retrieval.

With the exception of the early morning and evening, the AOD derived from the inversion of SSARA sky radiance measure-
5   ments is overestimated by sometimes more than 0.1, when compared with the values obtained from direct sun observations
from SSARA and AERONET (see Fig. 12). Additionally, the results from almucantar and principal plane differ significantly,
with neither of them preferable to the other. Judging from the residual, the results are all equally trustworthy, barring one
exception at approximately 13:00 UTC.





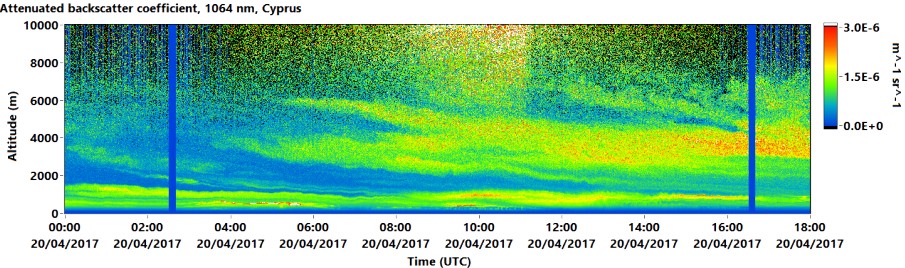

**Figure 11.** 1064 nm attenuated backscatter (in $m^{-1}sr^{-1}$) measured by a Polly$^{XT}$ lidar at the LACROS site on 20 April 2017.

An increase in the coarse mode AOD is clearly visible in Fig. 13b, starting at around 07:00 UTC. The retrieved values agree well with the AERONET inversion results. This increase is consistent with the arrival of Saharan dust which contains larger

5 particles. Consequently, the overestimation of the total AOD retrieved from SSARA sky radiance measurements has to be caused by the fine mode (see Fig. 13a). Also it is not consistently retrieved from principal plane and almucantar scan patterns. Again, the deviation in the retrieved total AOD from the direct sun observations is owed to the fact that this value is not used as a constraint in the inversion.

The effective radius of the fine mode (see Fig. 13c) is stable over most of the day, only increasing in the morning and

10 the evening again. AERONET finds larger fine mode particles, again by up to about 0.1 μm. Apart from the single outlier at 13:00 UTC, coarse mode effective radius is retrieved quite consistent over the entire day (Fig. 13d). Also, it agrees well with the AERONET inversion results. For values around 1.5 μm the retrieval proved to be reliable in the sensitivity studies. Here, an increase in morning and evening is visible as well.

For the real part of the refractive index (Fig. 13e and Fig. 13f) most measurements indicate a value of around 1.5 for both fine and coarse mode. This agrees well with the AERONET inversion, which produces only slightly lower values. However,

5 since this is also the prior and large discrepancies between values derived from the two scan patterns are visible, this might also be the result of lacking sensitivity to this parameter.

## 4    Summary & conclusions

The retrieval of microphysical and optical properties of aerosols from multispectral sky radiance observations remains a challenge, especially in cloudy conditions. Recently, the use of polarimetric information has proven to provide additional information. To use this, we introduce a new inversion method using such measurements. However, polarimetric measurements pose additional demands on the instruments, their setup and calibration. In this paper, we also present new methods to lower the effort of calibrating such an instrument and its mount. These methods are applicable to other instruments as well.

5 We introduced a new method for polarimetric calibration of polarized sun and sky radiometers. In contrast to previous calibration methods, it can simultaneously determine orientation and diattenuation of a polarized channel. This reduces the experimental effort, as only measurements at a single degree of polarization are necessary. Additionally, neither correction





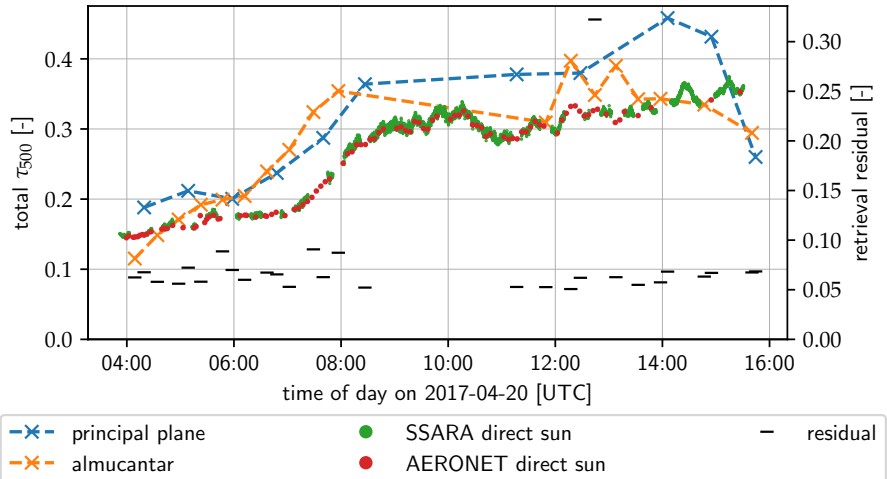

**Figure 12.** Same as Fig. 9, but for 20 April 2017.

factors nor assumptions about the filters are required. For the calibration of our sun photometer SSARA, the diattenuation of the linear polarizers was determined to an accuracy of 0.002, their rotation to within 0.1°. Neglecting these filter parameters would introduce a systematic relative error of up to 1.9 % in total radiance and 3.9 % in DoLP across the hemisphere.

A novel quaternion-based correction of the mount skewness reduces the pointing error of the instrument to below 32 arcmin. This is limited by the accuracy of SSARA's sun-tracker and could be improved with a more sophisticated one. The correction can either be applied in post-processing, reducing the demands on the accuracy of the setup of the mount. Alternatively, it can be used in real-time during the operation of the instrument, allowing for more precise pointing during cloudy days.

For evaluating our retrieval using polarimetric information, two days of SSARA measurements from the A-LIFE field campaign have been selected for more in-depth analysis. The SSARA instrument has been calibrated with the aforementioned methods. The retrieval has been applied on principal plane and almucantar scans separately. On both days, the results differ depending on the scan pattern used, the reason for which is not fully understood.

The first case study investigates the retrievals behaviour under partly cloudy conditions. An increase in AOD is visible around the time of convective activity. This effect has been shown to exist due to 3D radiative effects close to clouds in previous numerical studies. The second day selected features clear-sky conditions with an appearing Saharan dust layer. This layer can be observed by an increase in coarse mode AOD retrieved from SSARA measurements, as well as in AERONET inversion data. With a few exceptions, the retrieval shows the tendency to overestimate the AOD when compared to values obtained from direct sun observations. The error sometimes exceeds 0.1 in total AOD. The retrieval of the effective radius works well for the fine mode. In both cases, the value is slightly too low but agrees with AERONET to within 0.1 μm. In the coarse mode, the inversion compares well to AERONET for values around 1.5 μm. For larger particles (towards effective radii of 2 μm), our retrieval produces smaller radii than AERONET. There appears to be a systematic increase in the retrieved



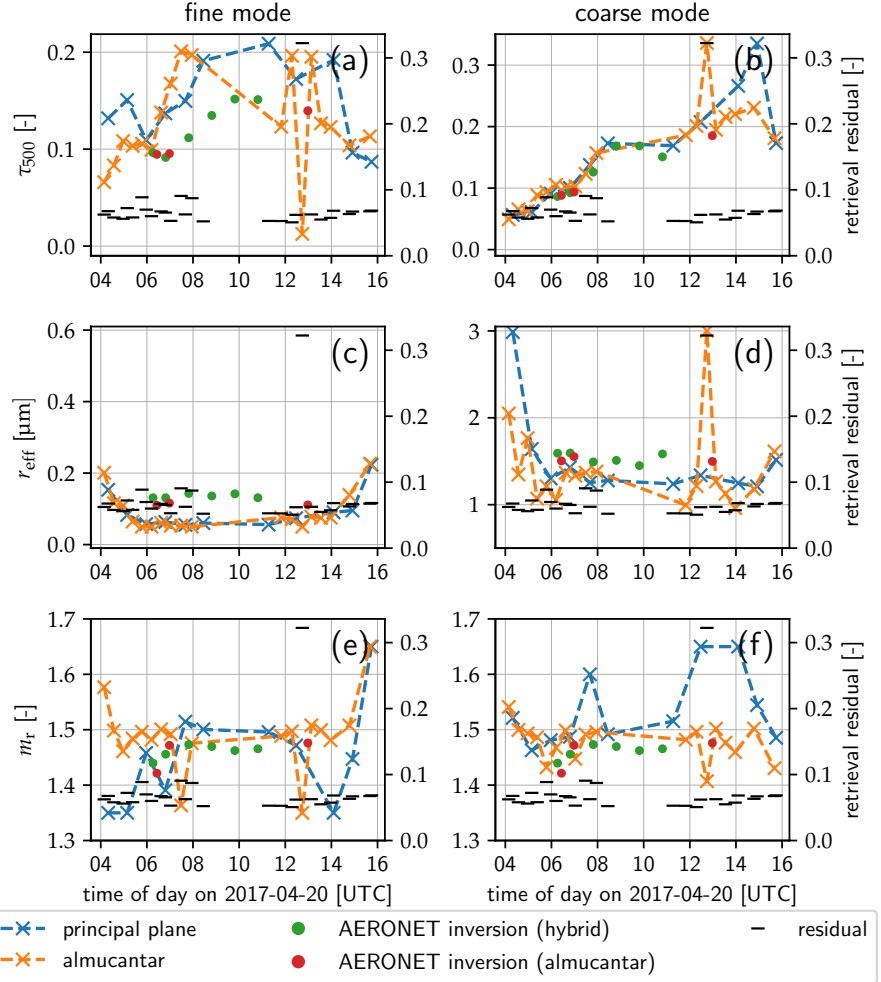

**Figure 13.** Same as Fig. 10, but for 20 April 2017.

effective radius for both modes in the morning and evening. To properly evaluate these results and resolve the remaining discrepancies, independent measurements are required. The same is true for the retrieval of index of refraction, especially due to the fact that AERONET uses a common value for both modes. In some cases, our results are well supported by AERONET. However, it often stays close to its prior, which could indicate lacking sensitivity to that parameter.

These remaining differences in the retrieved parameters between our method and the AERONET inversion have to be examined further. As a first step, the results from A-LIFE should be compared to measurements obtained from independent instruments, such as lidar or in situ. This should also extend to times where no AERONET results are available for comparison. Moreover, our inversion scheme should be applied to measurements from other sky radiometers, such as the Cimel CE318-DP used in AERONET. This is to rule out instrument effects. However, due to the high level of precision achieved in the





various calibration steps, this an unlikely source of error. Also, the retrieval could then be evaluated using multiple polarized wavelength measurements. Vice versa, our measurements might be analyzed using different inversion algorithms. This way systematic errors in the retrieval method can be identified. Further numerical studies with respect to the influence of the scan

pattern on the retrieval results are recommended. Additionally, it would be possible to add the total AOD obtained from direct sun observations as a constraint to our retrieval. This approach might limit the applicability to cloudy situations, when no such measurements are available or the value changes rapidly. However, for clear sky cases this constraint would certainly improve the retrieval results. Nonetheless, our polarimetric calibration method could easily be adapted to instruments used in AERONET.

*Code and data availability.* Code and data are made available on request.

## Appendix A: Introduction to Quaternions

Quaternions are an extension to complex numbers. As complex numbers can be used to describe operations - such as rotation - in 2D space (in polar notation), the same is true for quaternions in 3D space (see Horn, 1987). A quaternion is described by four real components,

$$\boldsymbol{q} = q_0 + iq_1 + jq_2 + kq_3 \,, \tag{A1}$$

where $i$, $j$, and $k$ are the imaginary units with the following identities,

$$i^2 = j^2 = k^2 = -1 \,,$$
$$ij = -ij = k \,, \quad jk = -kj = i \,, \quad ki = -ik = j \,. \tag{A2}$$

Quaternions form a non-Ablian group under multiplication defined by the *Hamilton product*. Therefore, quaternions do not

commute under the Hamilton product. It can be derived using the distributive and associative laws, and the identities in Eqs. (A2).

$$\boldsymbol{qu} = (q_0 + iq_1 + jq_2 + kq_3)(u_0 + iu_1 + ju_2 + ku_3) \tag{A3}$$
$$= + q_0u_0 + iq_0u_1 + ju_0q_2 + ku_0q_3$$
$$- q_1u_1 + iq_1u_0 - ju_1q_3 + ku_1q_2$$
$$- q_2u_2 + iq_2u_3 + ju_2q_0 - ku_2q_1$$
$$- q_3u_3 - iq_3u_2 + ju_3q_1 + ku_3q_0 \tag{A4}$$

Additionally, a dot product is defined as

$$\boldsymbol{q} \cdot \boldsymbol{u} = q_0u_0 + q_1u_1 + q_2u_2 + q_3u_3 \,. \tag{A5}$$





It can be used to induce a norm $\|\boldsymbol{q}\| = \sqrt{\boldsymbol{q} \cdot \boldsymbol{q}}$. A quaternion is conjugated by inverting the sign of its imaginary components,

$$\boldsymbol{q}^* = q_0 - iq_1 - jq_2 - kq_3 \,. \tag{A6}$$

It can be shown that the multiplicative inverse is

$$\boldsymbol{q}^{-1} = \frac{1}{\boldsymbol{q} \cdot \boldsymbol{q}} \boldsymbol{q}^* = \frac{1}{\|\boldsymbol{q}\|^2} \boldsymbol{q}^* \tag{A7}$$

As a result, for normed quaternions ($\|\boldsymbol{q}\| = 1$), its inverse is its conjugate.

Quaternions describing spatial rotations in three-dimensional space have to be normed. A rotation about an axis $\boldsymbol{a}$ through an angle $\alpha$ is represented by the quaternion

$$\boldsymbol{q}(\alpha, \boldsymbol{a}) = \cos\frac{\alpha}{2} + \sin\frac{\alpha}{2} \cdot (a_x i + a_y j + a_z k) \,. \tag{A8}$$

It can easily be seen, that the conjugate is in fact the inverse, corresponding to a rotation by the negative angle or around the negative axis. According to Euler's rotation theorem, the conjunction of several rotations can be described by a single rotation. This also follows from the group properties of quaternions. The Hamilton product of two normed quaternions is again a normed quaternion, representing a rotation.

A regular 3D Eucledian vector $\boldsymbol{r}$ can be described by a quaternion with a real part of $0$ (*pure* quaternion). The rotation by a quaternion is calculated as

$$\boldsymbol{r}' = \boldsymbol{q}\boldsymbol{r}\boldsymbol{q}^{-1} = \boldsymbol{q}\boldsymbol{r}\boldsymbol{q}^* \tag{A9}$$

The resulting quaternion is again pure, and the rotated vector can be reconstructed. Also, unit quaternions can be transformed into a rotation matrix that can be applied to regular Euclidean vectors. For a unit quaternion $q$, the Euler angles of the corresponding rotation and the $3 \times 3$ rotation matrix $\mathbf{M}_q$ are given by

$$\boldsymbol{r}' = \mathbf{M}_q \cdot \boldsymbol{r} \tag{A10}$$

$$= \begin{pmatrix} 1 - 2(q_2^2 + q_3^2) & 2(q_1 q_2 - q_0 q_3) & 2(q_1 q_3 + q_0 q_2) \\ 2(q_2 q_1 + q_0 q_3) & 1 - 2(q_1^2 + q_3^2) & 2(q_2 q_3 - q_0 q_1) \\ 2(q_3 q_1 - q_0 q_2) & 2(q_3 q_2 + q_0 q_1) & 1 - 2(q_1^2 + q_2^2) \end{pmatrix} \cdot \boldsymbol{r} \tag{A11}$$

*Author contributions.* HG developed the code for the retrieval and the calibration, performed the calibration, processed the measurement data and prepared the manuscript. MW and MS designed and built the SSARA instrument, respectively. CE and BM assisted the interpretation of the results. CE, MW, MS and BM also contributed to the manuscript. CE and BM prepared the proposal for the DFG project.

*Competing interests.* The authors declare that they have no conflict of interest.



*Acknowledgements.* The work for this paper was funded through the German Research Foundation (DFG) project 264269520 "Neue
Sichtweisen auf die Aerosol-Wolken-Strahlungs-Wechselwirkung mittels polarimetrischer und hyper-spektraler Messungen". We thank To-
bias Kölling and Markus Garhammer for their help with the calibration of the instrument. Carlos Toledano and his team operated and
maintained SSARA during most of the A-LIFE campaign. Linda Forster kindly provided the cameras mounted on the instrument, and the
corresponding software. Thanks to Holger Baars, Birgit Heese and the Polly[XT] team from the Leibniz Institute for Tropospheric Research
(TROPOS) in Leipzig, Germany, for performing the lidar measurements in Cyprus, creating the corresponding plot, and helping with its
interpretation. TROPOS acknowledges support from ACTRIS-2 under grant agreement no. 654109 from the European Union's Horizon
2020 research and innovation programme. The access to the LOA calibration facility, organized by Carlos Toledano, was possible thanks to
the ACTRIS project. An application of Transnational Access was approved by the AERONET-Europe panel within ACTRIS. Also thanks to
Maxime Catalfamo and Luc Blarel for their support at LOA. We thank Diofantos Hadjimitsis and his staff for their effort in establishing and
maintaining the CUT-TEPAK AERONET site.



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
