# Peer review of "The polarized sun and sky radiometer SSARA: design, calibration, and application for ground based aerosol remote sensing"

_Atmospheric Measurement Techniques, 2019_

## Referee Comment (RC1) · Gerard van Harten (Referee) · 24 Jul 2019

RECOMMENDATION:

Minor revisions

GENERAL COMMENTS:

The authors present the design, calibration, and application of the polarized sun and sky radiometer SSARA for ground based aerosol remote sensing. SSARA contains 12 radiometric channels within 340-1640 nm, similar to AERONET's CIMEL. Polarimetric capability was recently added to the $\sim$500 nm band. Four of the spectral bands, including the polarimetric band, are capable of performing diffuse sky measurements in addition to direct sun measurements, which are available for all bands. The magnitude and orientation of the diattenuation for the 3 polarimetric channels (0, -45, 90 degree analyzer orientations) are determined in the polarimetric calibration using the partially polarized POLBOX source. Radiative transfer simulations using MYSTIC show that the polarimetric calibration improves the radiometry by ∼1% and the degree of linear polarization by ∼2% (relative). Absolute radiometric calibration is performed in the lab using LOA's radiometrically calibrated SphereX, and on a mountain top using a Langley calibration. An advanced quaternion-based geometric calibration model was developed to derive the instrument pointing from direct sun views throughout the day. Recent modifications to the aerosol retrieval algorithm are described, followed by measurements and retrievals on a clear and a cloudy day during the A-LIFE campaign. On both days, the direct sun AOD matches the AERONET direct sun AOD closely, whereas the almucantar and principal plane retrievals typically overestimate the total AOD by ∼0.1. Separate AOD retrievals for fine and coarse mode show an increase in coarse mode which is linked to lidar measurements of incoming Sahara dust. In addition to AOD, the microphysical parameters of effective radius and real refractive index are retrieved for both size modes and compared to AERONET. The paper is very well written.

SPECIFIC COMMENTS:

1a. Section 4: "We introduced a new method for polarimetric calibration of polarized sun and sky radiometers. In contrast to pevious calibration methods, it can simultaneously determine orientation and diattenuation of a polarized channel. This reduces the experimental effort, as only measurements at a single degree of polarization are necessary." There are other mentions of the novelty of the polarimetric calibration as presented in the paper. However, polarimetric calibration using a single degree of polarization to simultaneously determine orientation and diattenuation of a polarized channel is common practice, typically using a rotating high-extinction polarizer.

1b. P5-14: "Since the angle of the plate can be determined with high accuracy, also the

DoLP is known to a high precision." Is the word "precision" used instead of "accuracy", because there is uncertainty in the refractive index of the glass plates? What is the total uncertainty in the POLBOX output DoLP?

1c. The POLBOX can generate DoLP between 0 and ∼58%. It seems like the polarimetric calibration only uses a single DoLP, i.e. the maximum DoLP. What is the advantage of using the POLBOX at a single DoLP of 58% vs a high-extinction polarizer which would provide the maximum possible calibration signal?

2a. The validation and interpretation of the aerosol retrieval results using the AERONET comparisons would probably benefit from having error bars on both the SSARA and AERONET retrievals. For example, the effective radii are somewhat different, but it is unclear how significant the differences are, and the refractive index seems to lack sensitivity as pointed out by the authors, but it does vary on the $\pm 0.1$ level. Another example is the coarse mode AOD during the Sahara dust event which matches the AERONET retrievals very closely, but the fine mode AOD is off in a seemingly systematic way.

2b. It is pointed out that the direct sun total AOD is never used as a constraint in the aerosol retrievals, in contrast to AERONET, because direct sun measurements might not be available in cloudy situations. To better understand the discrepancies in the retrieval results, it may be instructive to analyze the effect of using direct sun constraints whenever available.

DETAILED COMMENTS:

P2_2: "not good enough to properly resolve": What would be the required resolution?

P2_13-17: Since the paper is about a groundbased instrument, consider using the following references for GroundSPEX and GroundMSPI:

- Van Harten et al., AMT 7:4341-4351, 2014

- Di Noia et al., AMT 8:281-299, 2015

- Diner et al., Atmosphere 3:591-619, 2012

P3_4: "SSARA is . . . Munich": Provide reference, or is this it?

P3_13: "channels 13-15 are equipped with linear polarizers": Has this been published before, or is this a recent addition?

P3_14: "linear polarizers": What kind of polarizers?

P4_Table-1: Consider adding column "direct sun / diffuse sky"

P4_Table-1: Channels 4 and 13-15 are very similar. Have their radiances been compared?

P4_10: "simultaneously, because it does not use a filter wheel": How long does the AERONET filter wheel sequence take?

P5_6: Add "Li et al., 2018"

P5_22: "exiting light" -> "refracted light inside the glass"

P6_Figure-2: Consider adding dimensions and angle theta.

P7_8: "transmission values": amplitude or intensity transmission?

P7_26: "It is independent of the intensity of the incoming radiation I0". But it is retrieved from sequential measurements at different POLBOX angles, so light source stability is not unimportant.

P8_Table-2_caption: "unpolarized, so . . . D=0": It is not uncommon for non-polarimetric channels to exhibit diattenuation. The POLBOX experiment could be used to quantify and potentially correct polarization sensitivity in the non-polarimetric channels.

P14_9-13: Was the radiometric calibration on Mount Zugspitze compared to the radiometric calibration in the LOA lab using SphereX?

P15_8: "A ground albedo of 0.15 . . . is used for all wavelengths": Is that a good assumption? What kind of surface was assumed?

P16_25: "AOD data" -> "AOD data in Fig. 9"

P16_29: "as shown in sensitivity studies": Reference?

P17_5: "in space": How realistic is this explanation at the deployment location?

P17_6: "in time . . . 15 min": Figure 9 shows very stable direct sun AOD for hours.

P18_Figure-9: Don't forget to point out that the direct sun results match AERONET very well.

P20_bottom: "single degree of polarization": Was the POLBOX used at different DoLP settings, for example to validate the accuracy of the polarimetric calibration?

P21_10: "error of up to 1.9% in total radiance and 3.9% in DoLP" -> "error of up to $\pm1\%$ in total radiance and $\pm2\%$ in DoLP"

―――――――――――――――――――――

---

## Referee Comment (RC2) · Anonymous Referee #2 · 6 Oct 2019

GENERAL COMMENTS:

In this submission, the authors present the design, calibration and application of a polarisation sensitive radiometer (SSARA) for direct sun and skylight observations. The SSARA features 15 channels sensitive to different wavelength bands between 340 to 1650 nm. At 500 nm there are three channels additionally equipped with polarizing filters at different orientation. Measurement principle and design are very similar to other radiometers, e.g. the Cimel sun photometer used within the Aerosol Robotic Network (AERONET). The calibration procedures described involve polarimetric, radiometric and pointing calibration. The instrument's applicability in the field is demonstrated for

different conditions on data from two days of a measurement campaign conducted in 2017. In this context, the authors also apply a recently developed aerosol property retrieval algorithm. The obtained results are compared to data from a nearby located AERONET station.

I hold this publication to be of value for the scientific community, predominantly because it provides a well written and structured overview on the challenges and possible solutions encountered during the development of state of the art polarized radiometers. Further, it constitutes an important reference to cite in future project and/or instrument related publications. I recommend publication after the comments below have been adressed.

SPECIFIC COMMENTS:

I do not fully agree with the statement, that the presented approaches for the polarimetric and mount calibrations are "novel" or "new", since they mostly follow the procedures described in previous publications that the paper even refers to (Balois 1998 and Riesing 2018, respectively). In the case of the mount calibration I suggest to use "recently developed" instead. Regarding the polarimetric calibration: The approach is valid but I do not see the new/advantageous aspect here because: 1.) I expect that using a single DoLP makes the calibration less reliable, however for some applications this might be outweighed by the reduced effort. 2.) in my eyes using a single DoLP makes the POLBOX (with the main advantage to produce variable DoLPs) obsolete. Wouldn't it be much easier and probably even more accurate to replace the POLBOX by a polarizer with high extinction ratio (e.g. Glan-type polarizer: 10ˆ5) as it is often done?

The approach to use separate telescopes for each channel allows for simultaneous measurements of different wavelengths and polarisations. I would consider this as an advantage over filter wheel instruments in particular when it comes to aerosol observations during cloudy conditions with high temporal variability in intensity. This aspect
might be pointed out more clearly.

P9: Simulation of uncertainties with MYSTIC: Shown are the uncertainties for an uncalibrated instrument. It might be interesting to do the same simulation for the calibrated instrument (inserting the remaining uncertainties from table 2 into the RTM) to demonstrate the improvement and to see how accurate one can get applying the described calibration method.

Section 3: I think for a meaningful comparison it is important to add errorbars here for the AERONET and the SSARA direct sun observations. For the SSARA skylight measurements showing the retrieval residual is sufficient. However, also here an additional sentence in how far the residual reflects the real uncertainty of the respective values might be useful.

P2, Figure 1: It might be interesting to add one or two subfigures here from another perspective (e.g. showing the sensor head on the mount and the controller box respectively), giving the reader a better impression on the size and appearance of the total instrument setup. If this cannot be done I would suggest to at least add an approximate size indicator in the present figure.

P3, L8: Why was FOV=1.2° chosen? Sun disk + sun tracker inaccuracy?

P3, L14: What type of polarizers are used in the instrument? Wire-grid? Glan-Type?

P4, L6-8: I do not understand the details here. How/where is this baffle mounted? Onto the telescope shown in Image 1? Where do the 3.5° come from? And in general: which elements limit/define the FOV? Might be an option to add a sketch of one channel with the important optical elements and light path geometries.

P5, L14: "Known to a high precision". Don't you need the accuracy for the calibration? According to my experience measuring absolute angles accurately is not trivial. How well does that work?

P8, Table 2: Clearly for the unpolarised channels the diattenuation "D" should be zero

by theory. But has this been measured? If yes, I would suggest to insert and discuss the obtained values, as they may provide valuable information on the reliability of the instrument and/or the calibration method. Regarding the diattenuation of the polarized channels: I do not know what kind of polarisers are applied in the SSARA, but the values seem rather small. The same holds for the uncertainties. Considering that moving parts are used during the calibration procedure, are the uncertainties realistic? This is one of the reasons why the diattenuation values for the unpolarised channels mentioned above would be of interest. Finally, there are no uncertainties for a' given. They should be added if available.

TECHNICAL CORRECTIONS:

P5, L2: Typo: "Polarimetric calibration"

P5, L10: when introducing the DoLP here already, maybe add a reference to equation 10 on the next page.

P6, L14: "orders of magnitude smaller". How many? Just an approximate number would be nice.

P7, L21+22: remove "since" or "so" from the sentence.

P9, Figure 2 and P10, Figure 5: I suggest to indicate the sun's position by inserting a dot or sun symbol at the respective position in the polar plot.

P9, L13: Remove "by" from "varies by between" (?)

P12, L5: Should be "r_s" instead of "r_v" at end of line (?)

P12, L35: "using" instead of "used"

P13, L15: typo: "irrandiance"

P14, L23: Please reference Grob 2019 once more here. Otherwise it is not clear that the mentioned validation is published there.

P15 L10: "has been be revised": remove "be"

P16, L5: "To evaluate of the retrieval": remove "of"

P16, L33: Meaning of the black tickmarks might be moved to section before where the other general remarks on the plots are given.

P17, L15: At end of line: reference Grob 2019 again here.

P17, L18: "The same is true for the coarse mode ..." is misleading, since it is not clear to which statement in the sentence before "same" refers to: To "effective radii are somewhat smaller" (which is true) or to "within the 0.1 $\mu$ limit" (which is definitely not the case). Suggestion: "An underestimation is also observed for the coarse mode ..."

Figure 10 + 13: What is the "hybrid" AERONET inversion? Is it described somewhere?

P20, end of Section 3: "lacking sensitivity": Why would you expect that? From investigations in Grob 2019 or Xu 2015 maybe? If so, please add the corresponding reference.

P20, Section 4, L3: Remove "To use this,"

P21, L19: "retrieval's"

P23, L1: "this an unlikely error": add "is" here

P24, L17: "... a quaternion with a real part of 0": Changing this to "a quaternion with q_0 = 0 and q_1, q_2 and q_3 being the Eucledian vector components in x, y and z direction" might improve clarity here.

---

## Author Response (AR1)

Dear Udo Friess,

thank you very much for your quick and positive response. We addressed the issue you mentioned.

Sincerely,
Hans Grob

**Editor's comment**

Dear Hans Grob,

thank you very much for the submission of your manuscript on a polarized sun and sky radiometer. I feel that the manuscript is ready for entering the discussion phase after a minor technical correction regarding Figure 6, where East should be 90° clockwise from North, but is depicted in the opposite direction. Also, the U vector should point downwards in order to ensure a right-handed coordinate system (as it is correctly described in the text where z-axis points towards nadir, not zenith).

Best regards

Udo

*We corrected the definition of the ENU (east, north, up) coordinate system in Fig. 6, as well as the textual description in Sec. 2.2.2.*

Dear Referee,

thank you for your detailed review and valuable comments. We tried to address the issues you mentioned.

Sincerely,
Hans Grob

**Referee's specific comments**

1a. Section 4: "We introduced a new method for polarimetric calibration of polarized sun and sky radiometers. In contrast to pevious calibration methods, it can simultaneously determine orientation and diattenuation of a polarized channel. This reduces the experimental effort, as only measurements at a single degree of polarization are necessary." There are other mentions of the novelty of the polarimetric calibration as presented in the paper. However, polarimetric calibration using a single degree of polarization to simultaneously determine orientation and diattenuation of a polarized channel is common practice, typically using a rotating high–extinction polarizer.

*We agree that the calibration methods themselves are not novel. Yet, all literature currently being cited on the calibration of sun photometers (esp. AERONET) lack a rigorous mathematical derivation of the formalism and a description of the corresponding calibration process. This and the fact that diattenuation and polarizer angles can be determined in a single step are – in out opinion – actual improvements over the existing methods. Especially as this allows for the use of a less sophisticated polarized light source (see later).*
*However, "novel" is maybe a bit of an exaggeration in this case, so the wording was changed.*

1b. P5 ll. 14: "Since the angle of the plate can be determined with high accuracy, also the DoLP is known to a high precision." Is the word "precision" used instead of "accuracy", because there is uncertainty in the refractive index of the glass plates? What is the total uncertainty in the POLBOX output DoLP?

*We meant to say that the high accuracy of the DoLP from the POLBOX stems from that fact, that angles can (arguably) be determined rather precise, experimentally.*
*The wording was changed.*
*Additionally, the DoLP uncertainties from Li 2010, 2018 are cited. These are then used to determine the induced uncertainty in the calibration*

*of the diattenuation D.*

1c. The POLBOX can generate DoLP between 0 and ~58 %. It seems like the polarimetric calibration only uses a single DoLP, i.e. the maximum DoLP. What is the advantage of using the POLBOX at a single DoLP of 58 % vs a high–extinction polarizer which would provide the maximum possible calibration signal?

*Yes, this is true. In fact, POLBOX was as employed it was historically used for SSARA, and the people at Lille kindly granted us access. The possibility of setting an arbitrary DoLP is not needed in our method.*
*Also, as now mentioned in the text, the uncertainty of D seems to be limited by the uncertainty of the DoLP produced by the POLBOX. A polarizer with a higher precision would likely improve the calibration of D.*
*(Furthermore, it can be argued that a higher DoLP would improve the fit to Eq. (17) as it increases the amplitude of the cosine.)*

2a. The validation and interpretation of the aerosol retrieval results using the AERONET comparisons would probably benefit from having error bars on both the SSARA and AERONET retrievals. For example, the effective radii are somewhat different, but it is unclear how significant the differences are, and the refractive index seems to lack sensitivity as pointed out by the authors, but it does vary on the $\pm 0.1$ level. Another example is the coarse mode AOD during the Sahara dust event which matches the AERONET retrievals very closely, but the fine mode AOD is off in a seemingly systematic way.

*This is true. However, AERONET also does not provide any errorbars on their results. AERONET is currently working on a method to determine the errors based on perturbations of the measurement data. This is computationally very expensive. In the future, a similar approach could be applied to our retrieval. For now, the residual is shown. While this is by no means a proper uncertainty consideration, it is used as a proxy for the quality of the inversion.*

2b. It is pointed out that the direct sun total AOD is never used as a constraint in the aerosol retrievals, in contrast to AERONET, because direct sun measurements might not be available in cloudy situations. To better understand the discrepancies in the retrieval results, it may be instructive to analyze the effect of using direct sun constraints whenever available.

> *The decision not to include direct sun measurements in the retrieval process was consciously made early in the design process for the reasons mentioned in the paper. In hindsight, it would have been smarter to allow it, for exactly the cases you describe. Unfortunately, due to the structure of the code, this ability can hardly be added without changing the solver and other core components of the retrieval. Future retrieval designs should include this possibility.*
>
> *Still, while the statement that direct measurements would improve the retrieval results might seem somewhat intuitive (more information yields better results), it is purely speculative at this point and probably should not be made without prior studies. Therefore, we changed the wording of this statement.*

**Referee's detailed comments**

P2 ll. 2: "not good enough to properly resolve": What would be the required resolution?

> *Clarified and required resolution quantified.*

P2 lls. 13–17: Since the paper is about a groundbased instrument, consider using the following references for GroundSPEX and GroundMSPI:

- Van Harten et al., AMT 7:4341-4351, 2014

- Di Noia et al., AMT 8:281-299, 2015

- Diner et al., Atmosphere 3:591-619, 2012

> *References added.*

P3 ll. 4: "SSARA is...Munich": Provide reference, or is this it?
P3 ll. 13: "channels 13–15 are equipped with linear polarizers": Has this been published before, or is this a recent addition?

> *This is the first paper describing the instrument in depth. Additionally, SSARA took part in the SAMUM and SALTRACE campaigns, as mentioned in the introduction. The respective papers also contain information about the instrument. The polarized channels have been added in 2015 but not been used or described since.*

▌ P3 ll.14: "linear polarizers": What kind of polarizers?

> *Clarified. The filters are film sheet polarizers.*

▌ P4 Tab. 1: Consider adding column "direct sun/diffuse sky"

> *Column added to respective table.*

▌ P4 ll. 10: "simultaneously, because it does not use a filter wheel": How long does the AERONET filter wheel sequence take?

> *Clarified.*

▌ P5 ll. 6: Add "Li et al., 2018"

> *Done.*

▌ P5 ll. 22: "exiting light" → "refracted light inside the glass"

> *Changed.*

▌ P6 Fig. 2: Consider adding dimensions and angle theta.

> *Adding the angle in 2D is confusing. However, the caption has been updated in an effort to clarify the setup.*

▌ P7 ll. 8: "transmission values": amplitude or intensity transmission?

> *Clarified.*

▌ P7 ll. 26: "It is independent of the intensity of the incoming radiation $I_0$". But it is retrieved from sequential measurements at different POLBOX angles, so light source stability is not unimportant.

> *Clarified.*

P8 Tab. 2 (caption): "unpolarized, so...D=0": It is not uncommon for non–polarimetric channels to exhibit diattenuation. The POLBOX experiment could be used to quantify and potentially correct polarization sensitivity in the non–polarimetric channels.

*This is true and should be examined during the next calibration. However, due to time constraints we did not perform the necessary measurements of the unpolarized channels behind the POLBOX to characterize their diattenuation in this calibration session. The caption is changed to clarify $D = 0$ is an assumption and a short paragraph has been added as an explanation.*

P4 Tab. 1: Channels 4 and 13–15 are very similar. Have their radiances been compared?
P14 lls. 9–13: Was the radiometric calibration on Mount Zugspitze compared to the radiometric calibration in the LOA lab using SphereX?

*They have not been compared yet.*
TODO: this.

P15 ll. 8: "A ground albedo of 0.15...is used for all wavelengths": Is that a good assumption? What kind of surface was assumed?

*Clarified.*
TODO: this.

P16 ll. 25: "AOD data" → "AOD data in Fig. 9"

*Changed.*

P16 ll. 29: "as shown in sensitivity studies": Reference?

*Reference added.*

P17 ll. 5: "in space": How realistic is this explanation at the deployment location?

*The measurements site was close to the coast in Cyprus, where the different aerosol types described in the text could generally coincide. As a matter of fact, the presence of different aerosol types was one of the reasons*

*Cyprus was chosen as the location for the campaign. However, is this argument remains valid for the given days can only be speculated (see also next point).*

P17 ll. 6: "in time...15 min": Figure 9 shows very stable direct sun AOD for hours.

*This is true, so in this case, variation of AOD over the scanning time is probably not the explanation for the deviation. We removed the "scanning time" argument from the sentence.*

P18 Fig. 9: Dont forget to point out that the direct sun results match AERONET very well.

*Noted in the caption of Fig. 9 and the introduction to the case studies (Sect. 3.2).*

P20 (bottom): "single degree of polarization": Was the POLBOX used at different DoLP settings, for example to validate the accuracy of the polarimetric calibration?

*No, this has not been done yet. Ideally, the calibration would be performed using a high–performance polarizer, and check be done with POLBOX.*

P21 ll. 10: "error of up to 1.9 % in total radiance and 3.9 % in DoLP" → "error of up to ±1 % in total radiance and ±2 % in DoLP"

*Changed.*

Dear Referee,

thank you for your detailed review and valuable comments. We have addressed all the points you mentioned.

Sincerely,
Hans Grob

**Referee's specific comments**

> I do not fully agree with the statement, that the presented approaches for the polarimetric and mount calibrations are "novel" or "new", since they mostly follow the procedures described in previous publications that the paper even refers to (Balois 1998 and Riesing 2018, respectively). In the case of the mount calibration I suggest to use "recently developed" instead.

*As this point was also raised by reviewer #1 the same answer is provided:*
*We agree that the calibration methods themselves are not novel. However, all literature currently being cited on the calibration of sun photometers (esp. AERONET) lack a rigorous mathematical derivation of the formalism and a description of the corresponding calibration process. This and the fact that diattenuation and polarizer angles can be determined in a single step are – in our opinion – actual improvements over the existing methods. Especially as this allows for the use of a less sophisticated polarized light source (see later).*
*However, "novel" is not the correct expression in this case, so the wording has been changed as suggested.*

> Regarding the polarimetric calibration: The approach is valid but I do not see the new/advantageous aspect here because: 1.) I expect that using a single DoLP makes the calibration less reliable, however for some applications this might be outweighed by the reduced effort. 2.) in my eyes using a single DoLP makes the POLBOX (with the main advantage to produce variable DoLPs) obsolete. Wouldn't it be much easier and probably even more accurate to replace the POLBOX by a polarizer with high extinction ratio (e.g. Glan-type polarizer: 105) as it is often done?

*POLBOX was used for SSARA calibration, because LOA (Lille) kindly granted us access to it. The possibility of setting and arbitrary DoLP is indeed not needed in our method.*
*Also, as now mentioned in the text, the uncertainty of D is limited by the uncertainty of the DoLP produced by the POLBOX. A polarizer with a higher*

*precision would likely improve the calibration of D, but we did not have this available.*

The approach to use separate telescopes for each channel allows for simultaneous measurements of different wavelengths and polarisations. I would consider this as an advantage over filter wheel instruments in particular when it comes to aerosol observations during cloudy conditions with high temporal variability in intensity. This aspect might be pointed out more clearly.

*Thank you for this comment. We included the following sentence: "All channels are installed parallel to each other, allowing for simultaneous measurements at different wavelengths and polarizations. This is a big advantage in particular for aerosol observations during cloudy conditions with high temporal variability."*

P9: Simulation of uncertainties with MYSTIC: Shown are the uncertainties for an uncalibrated instrument. It might be interesting to do the same simulation for the calibrated instrument (inserting the remaining uncertainties from table 2 into the RTM) to demonstrate the improvement and to see how accurate one can get applying the described calibration method.

*Thank you for this suggestion. We did the uncertainty simulations for the calibrated instrument and included the results in the manuscript.*

Section 3: I think for a meaningful comparison it is important to add errorbars here for the AERONET and the SSARA direct sun observations. For the SSARA skylight measurements showing the retrieval residual is sufficient. However, also here an additional sentence in how far the residual reflects the real uncertainty of the respective values might be useful.

*The error of the retrieval of AOD from direct sun observations is very small, errorbars would not be visible on the scale of the figures. Therefore we take those values as the "reference" in clear sky conditions.*
*To clarify that the residual does not reflect the real uncertainties we included: "Since the total error of the retrieved values cannot easily be estimated, we show the residual of the minimization as an indicator of the performance of the retrieval for a given measurement."*

P2, Figure 1: It might be interesting to add one or two subfigures here from another perspective (e.g. showing the sensor head on the mount and the controller box respectively), giving the reader a better impression on the size

and appearance of the total instrument setup. If this cannot be done I would suggest to at least add an approximate size indicator in the present figure.

*We have included another photograph showing the SSARA instrument on the alt-azimuthal mount with straylight baffle.*

P3, L8: Why was FOV=1.2° chosen? Sun disk + sun tracker inaccuracy?

*A FOV of 1.2° was chosen for consistency with CIMEL sun-photometers. This information has been added to the text.*

P3, L14: What type of polarizers are used in the instrument? Wire-grid? Glan-Type?

*The instrument includes linear film polarizer sheets. This information has been added to the text.*

P4, L6-8: I do not understand the details here. How/where is this baffle mounted? Onto the telescope shown in Image 1? Where do the 3.5° come from? And in general: which elements limit/define the FOV? Might be an option to add a sketch of one channel with the important optical elements and light path geometries.

*We have included an additional figure showing how the straylight baffle is mounted on the telescope. Moreover, we have tried to clarify the description of the straylight baffle in the text.*

P5, L14: "Known to a high precision". Don't you need the accuracy for the calibration? According to my experience measuring absolute angles accurately is not trivial. How well does that work?

*We refer to Li et al (2010 and 2018) who have determined the accuracy of the DoLP of the POLBOX.*

P8, Table 2: Clearly for the unpolarised channels the diattenuation "D" should be zero by theory. But has this been measured? If yes, I would suggest to insert and discuss the obtained values, as they may provide valuable information on the reliability of the instrument and/or the calibration method. Regarding the diattenuation of the polarized channels: I do not know what kind of polarisers are applied in the SSARA, but the values seem

rather small. The same holds for the uncertainties. Considering that moving parts are used during the calibration procedure, are the uncertainties realistic? This is one of the reasons why the diattenuation values for the unpolarised channels mentioned above would be of interest. Finally, there are no uncertainties for a' given. They should be added if available.

*For the unpolarised channels the diattenuation has not been measured. We obtained the small values of D from the Levenberg Marquardt fit using Eq. 17 as model and we do not find any mistake here. The error of a' cannot be determined as long as the error of the intensity emitted by the SphereX is unknown.*

**Referee's Technical Corrections**

P5, L2: Typo: "Polarimetric calibration"

*Done.*

P5, L10: when introducing the DoLP here already, maybe add a reference to equation 10 on the next page.

*Moved definition of Stokes vector and degree of polarization to beginning of Section.*

P6, L14: "orders of magnitude smaller". How many? Just an approximate numberwould be nice.

*It is about three orders of magnitude smaller. This has been added to the text and further references with benchmark results including circular polarizations have been added.*

P7, L21+22: remove "since" or "so" from the sentence

*Removed "so".*

P9, Figure 2 and P10, Figure 5: I suggest to indicate the sun's position by inserting adot or sun symbol at the respective position in the polar plot.

> *We included the sun position as red marker in the figures.*

▌ P9, L13: Remove "by" from "varies by between" (?)

> *Done.*

▌ P12, L5: Should be "r$_s$" instead of "r$_v$" at end of line (?)

> *Corrected.*

▌ P12, L35: "using" instead of "used"

> *Corrected.*

▌ P13, L15: typo: "irrandiance"

> *Corrected.*

▌ P14, L23: Please reference Grob 2019 once more here. Otherwise it is not clear thatthe mentioned validation is published there.

> *Included.*

▌ P15 L10: "has been be revised": remove "be"

> *Corrected.*

▌ P16, L5: "To evaluate of the retrieval": remove "of"

> *Corrected.*

▌ P16, L33: Meaning of the black tickmarks might be moved to section before where the other general remarks on the plots are given.

> *Included explanation of black tickmarks where general remarks on plots are given.*

P17, L15: At end of line: reference Grob 2019 again here.

*Included reference.*

P17, L18: "The same is true for the coarse mode..." is misleading, since it is not clear to which statement in the sentence before "same" refers to: To "effective radii are somewhat smaller" (which is true) or to "within the $0.1\mu$ limit" (which is definitely not the case). Suggestion: "An underestimation is also observed for the coarse mode..."

*Thank you for the suggestion, included.*

Figure 10 + 13: What is the "hybrid" AERONET inversion? Is it described somewhere?

*Added reference Giles 2019 for the description of the "hybrid" inversion.*

P20, end of Section 3: "lacking sensitivity": Why would you expect that? From in-vestigations in Grob 2019 or Xu 2015 maybe? If so, please add the corresponding reference.

*We included the reference Grob 2019 here, were the same was observed.*

P20, Section 4, L3: Remove "To use this,"

*Done.*

P21, L19: "retrieval's"

*Corrected.*

P23, L1: "this an unlikely error": add "is" here

*Included.*

[revised manuscript text omitted]

with

$B_1 = 1.73759695,$      $C_1 = 0.01318870700 \, \mu m^2,$

$B_2 = 0.313747346,$      $C_2 = 0.0623068142 \, \mu m^2,$

$B_3 = 1.898781010,$      $C_3 = 155.2362900 \, \mu m^2.$

The POLBOX has a maximum tilt angle of $\alpha = 65°$. The resulting DoLP is roughly $58\%$ at the SSARA polarized wavelength of $501.5 \, nm$.

In the *Stokes–Müller* formalism, interactions with optical components or the atmosphere are described by left multiplication of the Stokes vector of the incoming radiation $\boldsymbol{S_{in}}$ with the appropriate real $4 \times 4$ Müller matrices $\mathbf{M_1}$ to  $\mathbf{M_n}$,

$$\boldsymbol{S_{out}} = \mathbf{M_n} \cdots \mathbf{M_1} \cdot \boldsymbol{S_{in}}. \tag{11}$$

In this context, a linear polarizer can be described as a linear diattenuator, meaning its attenuation differs for the two directions of polarization. The Müller matrix $\mathbf{LD}$ for a linear diattenuator rotated by an arbitrary angle $\vartheta$ is given in Bass et al. (2010) as

$$\mathbf{LD}(\vartheta) =$$
$$\frac{1}{2} \begin{pmatrix} a & b\cos(2\vartheta) & b\sin(2\vartheta) & 0 \\ b\cos(2\vartheta) & a\cos^2(2\vartheta) + c\sin^2(2\vartheta) & (a-c)\cos(2\vartheta)\sin(2\vartheta) & 0 \\ b\sin(2\vartheta) & (a-c)\cos(2\vartheta)\sin(2\vartheta) & a\sin^2(2\vartheta) + c\cos^2(2\vartheta) & 0 \\ 0 & 0 & 0 & c \end{pmatrix}, \tag{12}$$

with $a = k_0 + k_1$, $b = k_0 - k_1$, and $c = 2\sqrt{k_0 k_1}$. $k_0$ and $k_1$ are the intensity transmission values for the filter in the direction parallel and perpendicular to its orientation, respectively. $\vartheta$ is the angle between the polarization direction of the incoming radiation and the filter. Since a photodiode can only measure the total intensity of the light (first component of Stokes vector), the measurement operator $\langle M|$ projects only the first row of the matrix. Mathematically, it can be described as a transposed vector $(1,0,0,0)$

$$I = \langle M | \mathbf{LD} | \boldsymbol{S} \rangle \tag{13}$$
$$= \frac{1}{2} [a \cdot I_0 + b \cdot \cos(2 \cdot \Delta\vartheta) \cdot Q_0 + b \cdot \sin(2 \cdot \Delta\vartheta) \cdot U_0] \tag{14}$$

The light entering the instrument behind the POLBOX is taken to be polarized only in the positive $Q$ direction. This means the Stokes vector is given by $(I_0, \eta_{tot} I_0, 0, 0)^T$, with $\eta_{tot}$ again being the degree of linear polarization produced by the POLBOX. Also, the sensor has a certain radiometric response $C$, so the measurement vector becomes $\langle M| = (C, 0, 0, 0)$.

$$S(\vartheta) = \frac{C}{2}[a \cdot I_0 + b \cdot \cos(2(\vartheta - \vartheta_0)) \cdot \eta \cdot I_0] \tag{15}$$
$$= \frac{1}{2}[a' \cdot I_0 + b' \cdot \cos(2(\vartheta - \vartheta_0)) \cdot \eta \cdot I_0] \tag{16}$$

[revised manuscript text omitted]

rotation - in 2D space (in polar notation), the same is true for quaternions in 3D space (see Horn, 1987). A quaternion is described by four real components,

$$\boldsymbol{q} = q_0 + iq_1 + jq_2 + kq_3 \,, \tag{A1}$$

where $i$, $j$, and $k$ are the imaginary units with the following identities,

$$i^2 = j^2 = k^2 = -1 \,,$$
$$ij = -ij = k \,, \quad jk = -kj = i \,, \quad ki = -ik = j \,. \tag{A2}$$

Quaternions form a non-Ablian group under multiplication defined by the *Hamilton product*. Therefore, quaternions do not commute under the Hamilton product. It can be derived using the distributive and associative laws, and the identities in Eqs. (A2).

$$\boldsymbol{qu} = (q_0 + iq_1 + jq_2 + kq_3)(u_0 + iu_1 + ju_2 + ku_3) \tag{A3}$$
$$= + q_0 u_0 + iq_0 u_1 + ju_0 q_2 + ku_0 q_3$$
$$\quad - q_1 u_1 + iq_1 u_0 - ju_1 q_3 + ku_1 q_2$$
$$\quad - q_2 u_2 + iq_2 u_3 + ju_2 q_0 - ku_2 q_1$$

[revised manuscript text omitted]